# NSNQuant: A Double Normalization Approach for Calibration-Free Low-Bit Vector Quantization of KV Cache

**Donghyun Son**
Seoul National University
happydh1@snu.ac.kr

**Euntae Choi**
Seoul National University
euntae.choi175@gmail.com

**Sungjoo Yoo**[*]
Seoul National University
sungjoo.yoo@gmail.com

## Abstract

Large Language Model (LLM) inference is typically memory-intensive, especially when processing large batch sizes and long sequences, due to the large size of key-value (KV) cache. Vector Quantization (VQ) is recently adopted to alleviate this issue, but we find that the existing approach is susceptible to distribution shift due to its reliance on calibration datasets. To address this limitation, we introduce **NSNQuant**, a calibration-free Vector Quantization (VQ) technique designed for low-bit compression of the KV cache. By applying a three-step transformation—**1)** a token-wise normalization (**N**ormalize), **2)** a channel-wise centering (**S**hift), and **3)** a second token-wise normalization (**N**ormalize)—with Hadamard transform, NSNQuant effectively aligns the token distribution with the standard normal distribution. This alignment enables robust, calibration-free vector quantization using a single reusable codebook. Extensive experiments show that NSNQuant consistently outperforms prior methods in both 1-bit and 2-bit settings, offering strong generalization and up to $3\times$ throughput gain over full-precision baselines.

## 1 Introduction

Large language models (LLMs) have been widely adopted across various domains due to their strong generalization capabilities [1]. Recently, with the emerging trend of using LLMs to solve complex problems, their use has expanded into long-context scenarios such as long-context reasoning [40, 15] and retrieval-augmented generation (RAG) [24]. However, when processing long sequences, LLM inference requires a significant amount of memory and is memory-bound [22, 19].

One of the major causes is the large size of key-value (KV) cache, which linearly increases with the sequence length. To alleviate this issue, many studies have proposed approaches to compressing KV cache effectively, such as eviction [44, 25, 4, 32] and low-rank approximation [5, 26]. Among them, quantization has been one of the most widely adopted approaches. Previous studies [30, 19] analyze how outliers emerge in the KV cache and propose to quantize key and value along the channel dimension and token dimension, respectively. Recently, Coupled Quantization (CQ) [43] proposed to quantize multiple channels together, using centroids obtained from the calibration set. The idea of CQ is identical to the concept of vector quantization (VQ), where groups of values are jointly quantized into codebook indices. Utilizing the centroids as codebooks, CQ achieves state-of-the-art performance in diverse tasks, proving the effectiveness of VQ in KV cache quantization.

In our study, we observe that CQ fails to generalize in out-of-distribution (OOD) scenarios where the input distribution is different from that of the calibration set (in-distribution) [42]. For example, Figure 1a shows that while CQ excels in WikiText-2, it performs much worse in C4, since it is

---

[*]Corresponding author.

39th Conference on Neural Information Processing Systems (NeurIPS 2025).

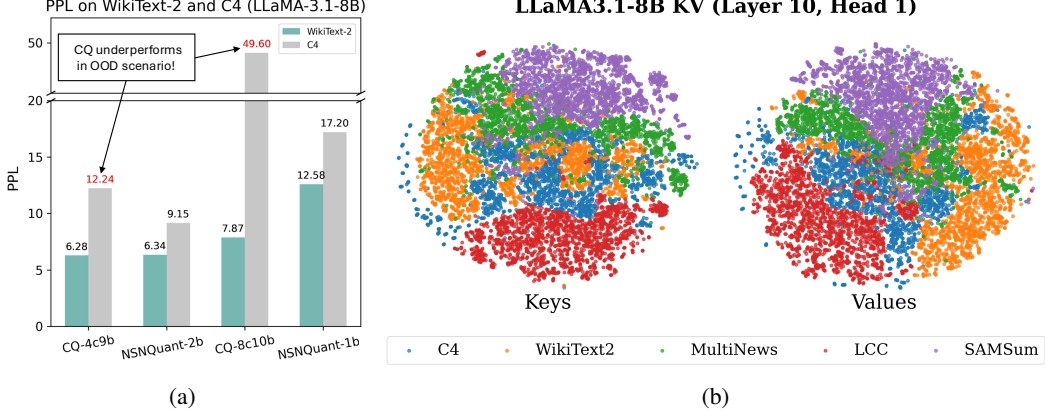

Figure 1: (a) PPL evaluation results with LLaMA3.1-8B. Although CQ achieves lower PPL in WikiText-2 (in-distribution), it performs worse on C4 (out-of-distribution). (b) t-SNE visualization of LLaMA3.1-8B key and value. The clustering pattern shows that key and value distributions strongly depend on the input data. More visualization can be found in Figure 11

calibrated using only a few samples from WikiText-2 dataset. Moreover, Figure 1b demonstrates the distribution of key and value varies greatly with the input distribution. This implies that learning centroids from the small set of data is very risky when applying them to other datasets. To this end, we propose NSNQuant, a calibration-free vector quantization (VQ) method that generalizes well to a wide range of datasets. Our contributions are summarized as follows:

- We empirically show that the strong dependence of key-value distributions on the input dataset can lead to severe errors in the existing VQ method. In particular, we observe that the centroids learned by CQ-4c9b fail to accurately quantize important punctuation tokens in LLaMA3-8B and LLaMA3.1-8B on C4, resulting in degraded performance.

- To address this limitation, we propose **NSNQuant**, a calibration-free VQ method for KV cache. NSNQuant effectively matches the key and value distribution with the standard normal distribution through the Normalize-Shift-Normalize (NSN) process and the Hadamard transform, as shown in Figure 3. Since the approximate distribution of key-value is known prior to inference, an effective codebook for VQ can be built without any external data.

- We conduct comprehensive experiments and analysis to show the effectiveness of NSNQuant across different tasks and models. The results with the LLaMA [31, 36, 14] and Mistral models [21] clearly show that NSNQuant outperforms other baselines in 1-bit and 2-bit quantization. Moreover, we implement efficient CUDA kernels for the low-bit computation, improving throughput and reducing memory usage.

## 2 Preliminary

**LLM inference and KV cache**   LLM inference has two main stages: prefilling and decoding. In the prefilling stage, all prompt tokens are processed simultaneously by the transformer decoder layers [38]. In the decoding stage, new tokens are generated one by one in an autoregressive manner. In both stages, each token only attends to previous tokens due to the causal nature of masked self-attention. To avoid redundant computation, a KV cache stores key-value pairs from previous tokens. It is initialized during prefilling and updated at each decoding step by appending the latest key-value pair. It becomes a primary bottleneck when processing long sequences, as its size linearly increases with the sequence length.

**Vector quantization**   Unlike scalar quantization (SQ) where each scalar value is quantized individually, vector quantization (VQ) compresses a group of values jointly using a codebook. In VQ, a $d$-dimensional vector is matched to the closest entry in a codebook, and quantized as follows:

$$\mathrm{VQ}(v) = \underset{i}{\arg\min}\, \mathrm{D}(v, \mathbb{C}[i]), \tag{1}$$

where $\mathbb{C}$ denotes the codebook and $\mathrm{D}(a, b)$ is a distance function between vectors $a$ and $b$.

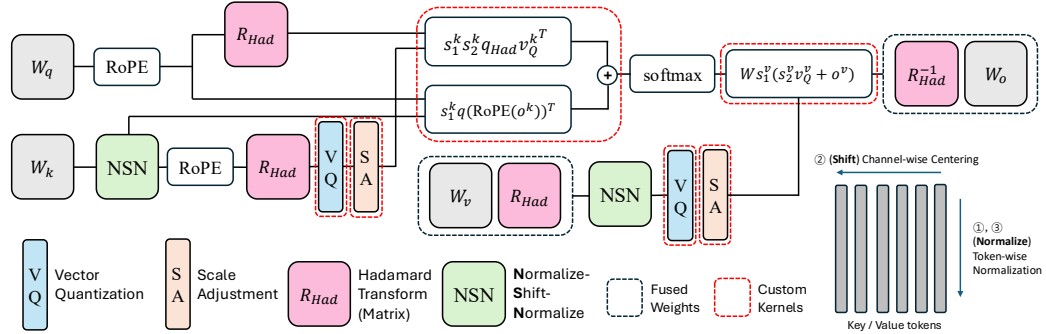

Figure 2: Overall structure of attention under NSNQuant. Residuals are omitted for simplicity. We use superscript $k$ and $v$ to mark values associated with key and value, respectively. Since NSN is applied to keys before RoPE, two branches are needed to correctly compute attention scores. Details of the attention computation shown in the figure are provided in Appendix B.

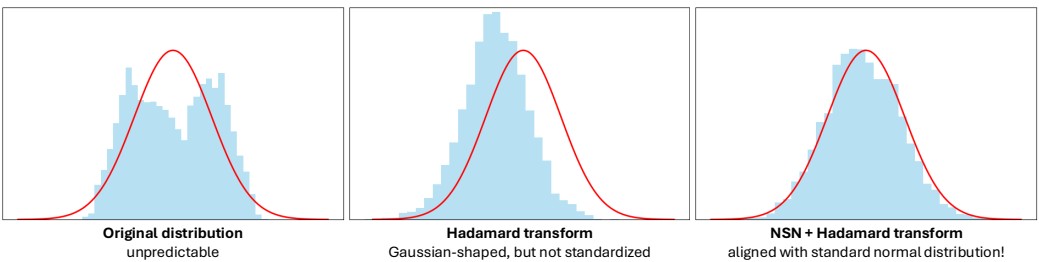

Figure 3: A visual illustration of the effect of NSN on the per-channel value distribution. Our Normalize-Shift-Normalize (NSN) process aligns the distribution with the standard normal distribution, when used with the Hadamard transform.

**Hadamard transform**   A Hadamard matrix is an orthogonal matrix whose entries all have the same magnitude. Its size can be doubled recursively through Sylvester's construction, which builds a larger Hadamard matrix by combining copies of a smaller one in a specific pattern. This recursive definition forms the basis of the Walsh–Hadamard transform, allowing matrix–vector multiplication in $O(d \log d)$ time. QuIP# [37] adopts the **randomized Hadamard transform (RHT)** [16], where the sign of each row and column is flipped independently with probability 1/2. We also employ RHT to compute the theoretical bounds presented in Lemma 1.

## 3  Method

### 3.1  Motivation

As shown in Figure 1a, we observe several cases where the existing VQ method, coupled quantization (CQ) [43], suffers from severe performance degradation when tested on OOD datasets. We attribute this to the distribution mismatch between datasets, as visualized in Figure 1b. We observe that this mismatch can cause severe errors in certain datasets. A notable example of such errors is the large quantization error observed in punctuation tokens in LLaMA3-8B and LLaMA3.1-8B.

WikiText-2, a calibration dataset for CQ, only contains " , " (with whitespace) tokens, while the C4 dataset includes "," (without whitespace) tokens. This mismatch leads to large errors for the keys corresponding to the punctuation token "," in the first layer of LLaMA3 models, since the centroids of CQ are obtained only from WikiText-2. This leads to significant distortion in attention weights because in certain heads, the "," token accounts for over 90% of the attention weights. By preserving the keys corresponding to these tokens in the first layer, the perplexity of CQ-4c9b on C4 is improved from 13.97 to 9.15 in LLaMA3-8B, and from 12.24 to 9.16 in LLaMA3.1-8B, closely matching the results of NSNQuant.

Table 1: Average channel-wise KL divergence with the standard normal distribution measured on key and value of LLaMA2-7B on WikiText-2 with sequence length of 4096. KL divergence is computed between the binned empirical distribution and the standard normal CDF. We used 4096 random samples for Oracle, to match the sequence length.

| Method | Key | | Value | |
|---|---|---|---|---|
| | w/ Had. | w/o Had. | w/ Had. | w/o Had. |
| N | 0.2476 | 0.5405 | 0.0817 | 0.1109 |
| NS | 0.3068 | 0.5261 | 0.0356 | 0.0666 |
| NSN | **0.0197** | **0.2230** | **0.0252** | **0.0537** |
| Oracle (torch.randn) | 0.0096 | | | |

To avoid these calibration-induced errors, we propose a calibration-free vector quantization (VQ) method, **NSNQuant**, which does not rely on any external data. While CQ tries to match the codebook to the key-value (KV) distribution, we instead propose to match the KV distribution to a well-known prior. Motivated by the recent success of Hadamard-based methods in producing normal-like output distributions regardless of the input, we introduce a novel transformation—Normalize–Shift–Normalize (NSN)—that aligns key and value channels to a standard normal distribution. This enables us to construct a single reusable codebook tailored for the standard normal distribution, making our method, NSNQuant, both calibration-free and robust across diverse inputs.

The overall structure of NSNQuant is depicted in Figure 2. Key-value tokens are first processed through the NSN and Hadamard transform. They are then compressed by vector quantization (VQ) using a codebook, and a scaling parameter $s_2$ is adjusted adaptively to the VQ result. The detailed process of NSN is described in Section 3.2. The post-scaling technique and the recipe for building a codebook are introduced in Section 3.3 and 3.4, respectively.

## 3.2 Normalize-Shift-Normalize (NSN)

NSN consists of three steps: **1)** a token-wise normalization (**N**ormalize), **2)** a channel-wise centering (**S**hift), **3)** a second token-wise normalization (**N**ormalize).

Let $v \in \mathbb{R}^{l \times d}$ be the tensor where $l$ is a sequence length and $d$ is a hidden dimension per head. In the first *Normalize* step, each token is normalized to have a norm of $\sqrt{d}$, where $d$ is a token dimension per head. This prevents outlier tokens [7] from dominating the next steps and incurring large errors in tokens with small magnitude. In the *Shift* step, channel-wise mean is calculated and subtracted, so that the resulting distribution is zero-centered. Finally, in the second *Normalize* step, each token is again normalized to have a norm $\sqrt{d}$. The entire process is formulated as follows:

1. **Normalize:** $s_1 \leftarrow \mathrm{norm}(v, \mathrm{dim=token})/\sqrt{d}, \quad v_{\mathrm{n}} \leftarrow v/s_1$
2. **Shift:** $o \leftarrow \mathrm{mean}(v_{\mathrm{n}}, \mathrm{dim=channel}), \quad v_{\mathrm{ns}} \leftarrow v_{\mathrm{n}} - o$
3. **Normalize:** $s_2 \leftarrow \mathrm{norm}(v_{\mathrm{ns}}, \mathrm{dim=token})/\sqrt{d}, \quad v_{\mathrm{nsn}} \leftarrow v_{\mathrm{ns}}/s_2$

Each step produces a byproduct—denoted as $s_1$, $o$, and $s_2$—which is used to restore the original tensor by $v = s_1(s_2 v_{\mathrm{nsn}} + o)$. Although the last step can deviate the channel-wise mean from zero, we find that its effect is negligible, as further discussed in Appendix D.1.

When used together with the subsequent Hadamard transform, our NSN process effectively aligns the channel distribution with the standard normal distribution, as shown in Table 1. ① As identified in previous studies [37, 39, 9, 20], the Hadamard transform results in a normal-like distribution, which is supported by the central limit theorem. ② NSN process roughly standardizes the distribution of each channel when used with the following Hadamard transform. Putting ① and ② together, the resulting channel distribution is aligned with the standard normal distribution. ② can be justified through the following lemma, which gives theoretical bounds for variances:

**Lemma 1.** *Let $X = (X_1, \ldots, X_d)^{\mathsf{T}} \in \mathbb{R}^d$ be the random vector from the joint distribution of the channels after the NSN process, which satisfies*

*(1) Nearly centered:* $\frac{1}{d} \sum_{i=1}^{d} (\mathbb{E}[X_i])^2 \leq \varepsilon,$    *(2) Normalized:* $\sum_{i=1}^{d} \mathbb{E}[X_i^2] = d,$

*(3) Covariance bound:* $\| \operatorname{Cov}(X) - \operatorname{diag}(\operatorname{Cov}(X)) \|_F \le \Gamma$.

*For the randomized Hadamard transform* $Y = \operatorname{RHT}(X)$, $0 < \alpha < 1$ *and* $i \in \{1, 2, \ldots, d\}$,

$$\operatorname{P}\Big( \operatorname{Var}(Y_i) \in \big[\, 1 - \varepsilon - \Gamma\,\beta_\alpha, \ 1 + \Gamma\,\beta_\alpha \,\big] \Big) \ \ge \ 1 - \alpha$$

*where* $\beta_\alpha := \dfrac{1}{d}\sqrt{\dfrac{\ln(2/\alpha)}{c}}$ *and* $c$ *is a constant from the Hanson-Wright inequality [34].*

The proof for the lemma is provided in Appendix A.1. Here, we adopt randomized Hadamard transform (RHT) to make the probabilistic claim, but using the naive Hadamard transform works well in practice, as presented in Table 7. Since NSN tightens the bound $\varepsilon$, and $\Gamma$ is small for most of the layers, we observe that the resulting variances are generally close to 1. However, we find that it does not hold in certain heads in the early layers, as presented in Figures 12 and 13. This is due to the presence of outlier channels with huge variances in the first layer, which enlarges the bound $\Gamma$. Despite this limitation, Figure 5 demonstrates that the quantization error remains low in these layers as well. We leave it to future work to explicitly model and account for these exceptions.

**Residual**     To implement the second step (Shift) in the decoding stage, we bring the idea of *residual* from KIVI [30]. Following KIVI, we split KV cache into two parts: one part with quantized KV cache, and the other part with full-precision KV cache (residual). If the size of the residual reaches its maximum capacity, the KV cache in the residual is flushed, quantized, and appended to the quantized part. We introduce a hyperparameter called *residual size* to control its size. To ensure consistency, we also apply NSN in residual-size chunks during the prefilling stage. In our experiments, we set the residual size to 64.

**NSN applied to key and value**     As illustrated in Figure 2, NSN is applied slightly differently to the key and value. For keys, NSN is applied immediately after the projection layer, and $v_{\text{nsn}}$ is quantized following the RoPE and Hadamard transform. While applying NSN after RoPE may seem more straightforward since RoPE might affect the channel-wise mean, we find that this ordering yields better quantization quality. Since RoPE is not applied to $o$ yet, we instead apply it within our custom kernel when computing attention scores, as shown in the figure. For values, the Hadamard transform is fused into the projection layers, and NSN is applied right afterward. Note that since the Hadamard transform is equivalent to multiplication by a rotation matrix, its order with the adjacent NSN does not change the output.

### 3.3 Scale adjustment

Let $v \in \mathrm{R}^d$ be a token vector that is processed by the NSN and Hadamard transform. It is then divided into 8-dimensional sub-vectors and quantized using the codebook, following the objective in Equation 1. Let $v_Q$ be the vector restored by looking up the codebook. i.e., $v_Q = \mathbb{C}[\operatorname{VQ}(v)]$. We find that rather than using $v_Q$ as-is, scaling $v_Q$ adaptively improves performance. Specifically, we find that it is beneficial to scale $v_Q$ as follows, which is identical to scaling $s_2$ since $v_Q$ is multiplied by $s_2$ when restoring:

$$v_Q \leftarrow \frac{\|v\|_2^2}{v \cdot v_Q} v_Q \quad (\text{i.e., } s_2 \leftarrow \frac{\|v\|_2^2}{v \cdot v_Q} s_2).$$

This is a scaling strategy which makes $v_Q - v$ orthogonal to $v$. In other words, it preserves components parallel to $v$, while allowing some orthogonal errors. Interpreting $o$ as local context and $v$ as a distinctive token feature, this strategy can be interpreted as making each token distinctive, which is essential for KV cache considering its selective property. We provide a comparison of different scaling strategies in Appendix C.3.

### 3.4 Codebook tuning

We construct a single global codebook for compressing 8-dimensional vectors using integer indices, following QuIP# [37]. NSNQuant-2b uses 8 bits for signs and 8 bits for codebook indices, while NSNQuant-1b uses only 8 bits for indices. A simple baseline can be built via K-Means on standard normal data, but its local optimality limits performance. We improve this by fine-tuning on synthetic

standard normal data (torch.randn) to minimize cosine distance between original and quantized vectors, since the error of scale adjustment depends only on the angle between them. As the lookup is non-differentiable, gradients are propagated only through post-lookup operations. The PyTorch implementation for this process runs in less than 5 minutes on an RTX 3090, unlike the calibration process of CQ or KVQuant which requires backpropagation through model weights. While E8P [37] is also a competitive 2-bit baseline, it is hard to apply to the 1-bit scenario, and requires expensive comparisons with over 2000 entries. We provide comparison results in Table 10.

### 3.5 Double quantization

To further reduce the memory overhead, we employ double quantization (DQ) proposed in QLoRA [10], which quantizes parameters used for the quantization. Specifically, we quantize $o$ and $s_1$ with a group size of 32 and *residual size*, respectively, using 4-bit round-to-nearest (RTN) quantization. DQ significantly reduces the average bit width. As a result, NSNQuant costs only additional 0.23 bits on average for the NSN process, when residual size is set to 64. Moreover, to reduce the amount of shared memory required for the codebook, we also apply 4-bit quantization to codebook entries. As shown in Table 12, DQ barely affects the effectiveness of NSNQuant.

### 3.6 Efficient kernel implementation

We implement the CUDA kernels for efficient execution of NSNQuant-2b and NSNQuant-1b. For codebook matching, we compute and manage distances on-the-fly in streaming multiprocessors (SMs), while loading codebook tiles into shared memory. For dequantization and matrix-vector multiplications, our kernel loads the codebook into shared memory to minimize access to DRAM. We implement two different matrix-vector multiplication kernels for $qK^T$ and $Wv$ ($W$: attention weights) since they use different axes for reduction. We also fuse the computation of both the quantized and the residual parts to maximize GPU utilization even in the small-batch scenarios.

## 4 Experiments

### 4.1 Experimental setup

**Datasets**  First, we evaluate perplexity (PPL) on WikiText-2 and C4 dataset to show quantization error in language modeling. Second, we report evaluation results on LongBench [3]. We choose the task subset from LongBench following KIVI [30]: Qasper for a single-document QA task; QMSum and MultiNews for summarization tasks; TREC, TriviaQA and SAMSum for few-shot learning tasks; LCC and RepoBench-P for code completion tasks. Lastly, we run evaluations on GSM8K [8], HumanEval [6], CoQA [33], and MMLU [18] using LM-Eval framework [13] to test generation capability in more diverse generation scenarios.

**Baselines**  We compare NSNQuant against four baselines: KIVI [30], KIVI + Hadamard, KVQuant [19], and CQ [43]. While some of these methods provide multiple versions with similar average bits (e.g., CQ-4c8b and CQ-4c9b), we adopt the stronger variants to better demonstrate the superiority of our approach. Simpler baselines like INT2 are excluded, as prior studies [19, 43] have shown them to be ineffective. For NSNQuant, we fix the residual size to 64. For KIVI, we use a group size of 64 for keys and 128 for tokens, to maintain consistency with our residual policy. Detailed explanations for each baseline are provided in Appendix E.3. Note that since CQ does not provide their official implementation, we reproduce it based on the official implementation of KVQuant. The result of CQ is slightly worse than the one reported in the original paper.

**Policy for full-precision cache**  Since maintaining full-precision cache largely affects generation quality [30, 43], we unify several settings to focus only on quantization quality: all methods adopt NSNQuant's residual policy, which buffers full-precision caches until the residual is full. We fix the residual size to 64 in all experiments. Attention sink-aware quantization is removed from KVQuant, as it can be easily applied to other methods as well. Since some methods (e.g., KIVI) use full-precision cache in the prefilling stage, while others (e.g., CQ) use quantized cache, we standardize this by using full-precision cache in the prefilling stage for all methods—except in perplexity evaluations, where key and value quantization is necessary to measure forward-pass quality.

## 4.2 PPL evaluation

Table 2: Perplexity (PPL) evaluation results on WikiText-2 and C4 with a context length of 4096. The results of CQ reported in the original paper are marked with †.

| Method | Avg. bit width | Dataset | LLaMA2-7B | LLaMA2-13B | LLaMA3-8B | LLaMA3.1-8B | Mistral-7B-v0.3 |
|---|---|---|---|---|---|---|---|
| FP16 | 16 | C4 | 6.63 | 6.04 | 8.32 | 8.43 | 7.48 |
|  |  | WikiText-2 | 5.12 | 4.57 | 5.75 | 5.84 | 4.95 |
| KIVI-2 | 2.38 | C4 | 8.00 | 7.03 | 16.43 | 15.80 | 8.83 |
|  |  | WikiText-2 | 6.14 | 5.30 | 10.93 | 10.55 | 6.03 |
| KIVI-2 + Had | 2.38 | C4 | 7.57 | 6.68 | 12.67 | 12.54 | 8.43 |
|  |  | WikiText-2 | 5.79 | 5.05 | 8.69 | 8.86 | 5.65 |
| KVQuant-2b + 1% | 2.32 | C4 | 7.09 | 6.37 | 9.75 | 9.60 | 7.93 |
|  |  | WikiText-2 | 5.52 | 4.88 | 6.74 | 6.71 | 5.32 |
| CQ-4c9b | 2.26 | C4 | 7.12 (7.02†) | 6.45 (6.36†) | 13.97 | 12.24 | 7.86 |
|  |  | WikiText-2 | 5.36 (5.32†) | 4.76 (4.74†) | **6.16** | **6.28** | 5.16 |
| NSNQuant-2b | 2.23 | C4 | **6.86** | **6.21** | **9.08** | **9.15** | **7.69** |
|  |  | WikiText-2 | **5.29** | **4.71** | 6.23 | 6.34 | **5.12** |
| KVQuant-1b + 1% | 1.32 | C4 | 30.79 | 14.27 | 33.17 | 37.37 | 12.45 |
|  |  | WikiText-2 | 13.5 | 9.91 | 27.57 | 33.96 | 9.06 |
| CQ-8c10b | 1.27 | C4 | 9.25 (9.12†) | 8.17 (8.01†) | 43.78 | 49.60 | 9.60 |
|  |  | WikiText-2 | 6.33 (**6.25**†) | 5.53 (**5.47**†) | 7.69 | 7.87 | 6.01 |
| NSNQuant-1b | 1.23 | C4 | **8.70** | **7.55** | **16.69** | **17.20** | 9.67 |
|  |  | WikiText-2 | 6.69 | 5.70 | 11.70 | 12.58 | 6.66 |

Table 2 presents the perplexity (PPL) of various models evaluated on WikiText-2 and C4. Although our framework uses full-precision key and value during the prefilling stage, we follow the evaluation protocol of CQ and KVQuant by quantizing them all in this experiment. Despite its difference from the generation scenario, we find PPL evaluation useful because it measures quantization quality for a sequence with a single forward pass. Furthermore, the results are aligned with those in Table 17, where we report the PPL evaluation results in the generation scenario with residuals. Therefore, we report PPL results in the main table, and use them in ablation studies.

Across both datasets, CQ and NSNQuant outperform the other methods. On WikiText-2, the two methods exhibit comparable performance in 2-bit quantization, with CQ outperforming NSNQuant in the 1-bit setting. However, on C4, NSNQuant consistently outperforms CQ in both 2-bit and 1-bit quantization. Notably, CQ suffers from severe performance degradation on C4 when using LLaMA3-8B and LLaMA3.1-8B, whereas NSNQuant maintains strong performance. These results suggest that NSNQuant generalizes better across datasets, while CQ struggles under distribution shift from the calibration dataset.

## 4.3 LongBench evaluation

Table 3: Evaluation results on LongBench. The task subset is selected following KIVI [30]. More results with different models can be found in Table 18.

| Model | Method | Bits | Qasper | QMSum | MultiNews | TREC | TriviaQA | SAMSum | LCC | RepoBench-P | Avg. |
|---|---|---|---|---|---|---|---|---|---|---|---|
| LLaMA3.1-8B-Instruct | FP16 | 16 | 13.11 | 23.53 | 26.74 | 72.50 | 91.65 | 43.78 | 63.04 | 56.17 | 48.82 |
|  | KIVI-2 | 2.38 | 12.04 | 24.96 | 26.70 | 72.00 | 91.97 | 43.43 | 60.85 | 53.39 | 48.17 |
|  | KIVI-2 + Had | 2.38 | 11.57 | 24.28 | 26.51 | 72.50 | 92.09 | 43.21 | 62.90 | 55.20 | 48.53 |
|  | KVQuant-2b + 1% | 2.32 | 13.15 | 23.45 | 26.24 | 72.00 | 91.63 | 41.39 | 60.80 | 54.41 | 47.88 |
|  | CQ-4c9b | 2.26 | 12.25 | 23.80 | 25.74 | 71.50 | 91.53 | 41.96 | 61.18 | 54.46 | 47.80 |
|  | NSNQuant-2b | 2.23 | 12.44 | 23.74 | 26.95 | 72.50 | 91.73 | 44.01 | 62.05 | 55.09 | **48.56** |
|  | KVQuant-1b + 1% | 1.32 | 9.91 | 22.19 | 22.27 | 47.50 | 88.92 | 35.76 | 50.27 | 43.79 | 40.08 |
|  | CQ-8c10b | 1.27 | 8.84 | 21.18 | 22.40 | 47.50 | 87.94 | 38.86 | 45.73 | 43.79 | 40.78 |
|  | NSNQuant-1b | 1.23 | 11.54 | 24.69 | 27.16 | 71.50 | 92.04 | 42.36 | 60.08 | 49.70 | **47.38** |
| Mistral-7B-Instruct-v0.3 | FP16 | 16 | 41.13 | 25.75 | 27.78 | 76.00 | 88.59 | 47.47 | 59.52 | 60.64 | 53.36 |
|  | KIVI-2 | 2.38 | 37.86 | 24.62 | 26.85 | 76.00 | 88.51 | 45.93 | 58.72 | 57.87 | 52.05 |
|  | KIVI-2 + Had | 2.38 | 39.99 | 25.42 | 27.50 | 76.00 | 88.42 | 46.52 | 59.54 | 60.13 | **52.94** |
|  | KVQuant-2b + 1% | 2.32 | 38.98 | 25.10 | 27.22 | 76.00 | 89.02 | 45.27 | 58.57 | 61.59 | 52.72 |
|  | CQ-4c9b | 2.26 | 39.85 | 24.50 | 27.19 | 76.00 | 88.86 | 45.56 | 58.36 | 60.26 | 52.57 |
|  | NSNQuant-2b | 2.23 | 39.96 | 24.91 | 27.54 | 76.00 | 88.96 | 46.47 | 58.70 | 59.45 | 52.75 |
|  | KVQuant-1b + 1% | 1.32 | 28.58 | 21.89 | 22.76 | 50.50 | 87.75 | 39.62 | 54.73 | 54.46 | 45.04 |
|  | CQ-8c10b | 1.27 | 31.21 | 22.56 | 23.12 | 64.50 | 88.09 | 41.71 | 53.82 | 52.31 | 47.16 |
|  | NSNQuant-1b | 1.23 | 37.94 | 25.03 | 26.81 | 76.00 | 89.39 | 46.37 | 56.75 | 55.57 | **51.73** |

Table 3 gives performance comparison on LongBench. In 1-bit quantization, NSNQuant-1b outperforms other baselines by a large margin. However, in 2-bit quantization, all methods exhibit similar performances. We attribute this trend to the noisy nature of certain tasks. For example, in code generation tasks like LCC and RepoBench-P, models tend to generate additional descriptions of the codes. These descriptions do not hurt the code quality, but eventually degrade the metric. To this end, for tasks with potentially noisy metrics, we also report the ROUGE-L score with the FP16 output to measure how well each method preserves the original model output. Table 19 clearly shows that NSNQuant best preserves the original model outputs.

## 4.4 Evaluation on more datasets

Table 4: Evaluation results on GSM8K, HumanEval, CoQA, and MMLU. Accuracy is reported for all tasks. Results with LLaMA2-13B-Chat and LLaMA3-8B-Instruct can be found in Table 20

| Model | Method | Bits | GSM8K (8-shot, CoT) | HumanEval | CoQA | MMLU (4-shot, CoT) | | | |
|---|---|---|---|---|---|---|---|---|---|
| | | | | | | Humanities | STEM | Social | Other |
| LLaMA3.1-8B-Instruct | FP16 | 16 | 76.65 | 57.93 | 63.78 | 71.47 | 57.96 | 74.16 | 72.52 |
| | KIVI-2 | 2.38 | 64.59 | 48.17 | 63.60 | 64.44 | 50.09 | 66.84 | 66.13 |
| | KIVI-2 + Had | 2.38 | 65.73 | 50.61 | **63.88** | 67.73 | 53.03 | 68.50 | 68.14 |
| | KVQuant-2b + 1% | 2.32 | 70.05 | 53.05 | 62.37 | 68.18 | 54.78 | 71.31 | 69.61 |
| | CQ-4c9b | 2.26 | 72.93 | 48.78 | 62.93 | 67.07 | 52.66 | 70.22 | 69.33 |
| | NSNQuant-2b | 2.23 | **75.89** | **56.10** | 63.83 | **71.04** | **55.64** | **73.42** | **70.74** |
| | KVQuant-1b + 1% | 1.32 | 21.53 | 23.17 | 53.55 | 23.04 | 11.23 | 37.59 | 33.02 |
| | CQ-8c10b | 1.27 | 44.88 | 25.61 | 56.58 | 28.21 | 21.34 | 31.97 | 41.10 |
| | NSNQuant-1b | 1.23 | **53.45** | **44.51** | **62.70** | **59.82** | **45.83** | **65.34** | **63.77** |
| Mistral-7B-Instruct-v0.3 | FP16 | 16 | 53.15 | 31.10 | 65.58 | 65.98 | 50.46 | 71.06 | 68.26 |
| | KIVI-2 | 2.38 | 43.75 | 28.66 | 64.45 | 60.96 | 39.93 | 63.52 | 59.05 |
| | KIVI-2 + Had | 2.38 | 46.10 | 28.05 | 65.48 | 63.28 | 45.11 | 66.99 | 63.07 |
| | KVQuant-2b + 1% | 2.32 | 46.63 | 27.44 | 64.28 | 63.00 | 45.47 | 67.39 | 66.27 |
| | CQ-4c9b | 2.26 | 47.84 | **31.10** | 64.80 | 62.48 | 42.48 | 68.73 | 63.98 |
| | NSNQuant-2b | 2.23 | **51.02** | **31.10** | **65.62** | **64.92** | **47.65** | **69.11** | **67.56** |
| | KVQuant-1b + 1% | 1.32 | 16.30 | 19.51 | 55.95 | 16.48 | 9.88 | 17.18 | 14.21 |
| | CQ-8c10b | 1.27 | 25.93 | 21.95 | 59.07 | 23.77 | 17.62 | 27.09 | 19.78 |
| | NSNQuant-1b | 1.23 | **38.89** | **27.44** | **63.60** | **58.52** | **40.34** | **62.34** | **58.43** |

In addition to the LongBench results, we further evaluate the models on GSM8K (mathematical reasoning), HumanEval (code generation), CoQA (conversational question answering), and MMLU (multi-task language understanding) to provide a more comprehensive assessment of generation quality. As shown in Table 4, NSNQuant outperforms other baselines in most of the settings. In particular, NSNQuant excels in GSM8K and MMLU in a few-shot CoT (Chain-of-Thought) setting, which requires models to generate an exact and strict reasoning path.

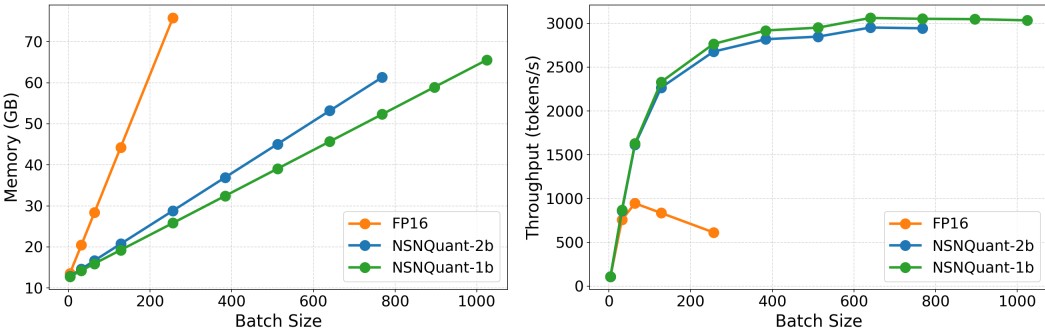

Figure 4: Peak memory usage (left) and throughput (right) measured with varying batch sizes. The residual size is set to 64. Results with varying residual sizes are available in Figure 8.

## 4.5 Memory usage and throughput

To evaluate the efficiency of NSNQuant in terms of both memory and runtime, we measure its memory consumption and throughput. Following previous works [30, 23], we use the synthetic data to simulate the scenario of ShareGPT [35] where the average input length is 161 and the average generation length is 338. The evaluation is performed on a Linux server with a single A100-80GB GPU with LLaMA2-7B.

Table 5: Latency (ms) breakdown of NSNQuant-2b compared with the FP16 baseline. HT and VQ refer to Hadamard transform and vector quantization, respectively. More results with different configurations can be found in Table 21

| Stage | Method | Total | NSN | HT | VQ |
|---|---|---|---|---|---|
| Prefill | FP16 | **1225.90** | – | – | – |
| | NSNQuant-2b | 1707.22 | 155.22 | 15.04 | 326.85 |
| Decode (per step) | FP16 | 50.80 | – | – | – |
| | NSNQuant-2b | **36.49** | 0.34 | 1.88 | 0.66 |

Table 6: PPL measured on WikiText-2 with LLaMA2-7B without each component of NSNQuant-2b.

| Method | PPL |
|---|---|
| NSNQuant-2b | **5.285** |
| w/o first token-wise normalization | 6.293 |
| w/o channel-wise centering | 5.842 |
| w/o second token-wise normalization | 5.456 |
| w/o Hadamard transform | 5.730 |

The result is illustrated in Figure 4. While FP16 baseline suffers from OOM for large batch sizes, NSNQuant-2b and NSNQuant-1b scale efficiently to larger batch sizes, achieving $4\times$ larger batch sizes and $3\times$ speedup in throughput. Note that the memory increases are not proportional to average bits since the residual is also included in the memory usage.

## 4.6 Latency breakdown

Although NSNQuant enables the use of larger batch sizes, it introduces additional computations for the Hadamard transform, vector quantization, and NSN. To better understand their impact on efficiency, we measure the latency of each operation during the forward pass. We set the batch size to 32 and the input length to 512, and generate 64 tokens to match the residual size. The latency is measured with LLaMA2-7B on a Linux server with A100-80GB, and the reported values are averaged over 100 runs.

The result is given in Table 5. Due to the additional overhead, NSNQuant has higher latency in the prefilling stage. On the other hand, in the decoding stage, it achieves lower latency since NSNQuant alleviates the memory-bound nature of the attention computation by compressing the KV cache into lower bits. This suggests that NSNQuant is particularly advantageous for decode-heavy tasks such as reasoning or code generation.

## 4.7 Ablation study

**Effects of the NSN and Hadamard**   To assess the contribution of each NSN component and the Hadamard transform, we measure perplexity on WikiText-2 while removing them individually. As shown in Table 6, skipping any step results in higher perplexity. This trend aligns with Table 1, which shows using NSN and the Hadamard transform together leads to the best alignment. The first token-wise normalization has the greatest impact, as it suppresses outlier tokens before centering. In contrast, the second token-wise normalization has minimal effect, suggesting that channel-wise centering does not significantly alter token scale.

**Effects of codebook tuning**   Figure 5 shows the cosine similarity between the original vector $v$ and the quantized vector $v_Q := \mathbb{C}[\text{VQ}(v)]$ with different codebooks. The fine-tuned version gives higher cosine similarities compared to the K-Means codebook. Furthermore, the cosine similarities are consistently high in the early layers, although the standardization of NSN does not hold in these layers. Notably, the cosine similarities measured using the key-value tokens (lines with markers) are very similar to those measured using synthetic standard normal data (dotted lines). This suggests that NSN successfully aligns the output distribution with the standard normal distribution, and the codebook trained only on synthetic data effectively quantizes such outcomes. As a result, codebook tuning improves perplexity from 5.294 to 5.285 with NSNQuant-2b and 6.910 to 6.703 with NSNQuant-1b.

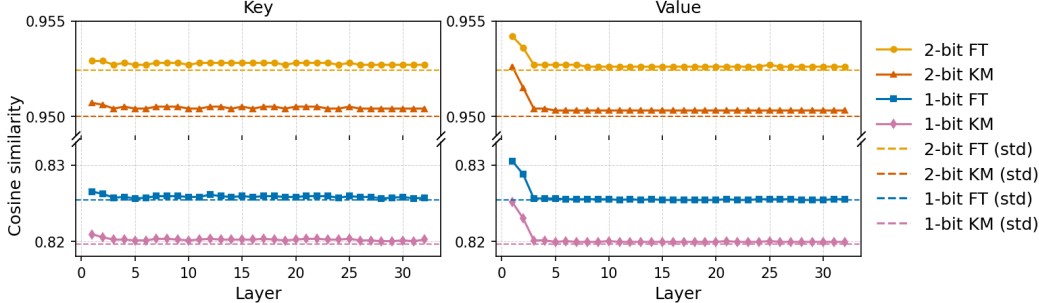

Figure 5: Cosine similarity between original and reconstructed vectors when applying VQ to the NSN-processed keys and values of LLaMA2-7B. Similarities are measured on WikiText-2 dataset with different codebooks. KM denotes the K-Means codebook, and FT denotes the fine-tuned codebook. The lines with markers show cosine similarity measured using the key-value from the model, whereas the dotted lines show measurements using synthetic standard normal data.

## 5  Related work

Recent efforts to reduce the memory and latency bottlenecks of LLMs have largely focused on weight quantization and KV cache compression. Weight quantization methods such as GPTQ [12] and AWQ [27] significantly reduce the memory footprint of model weights while preserving accuracy. In particular, vector quantization (VQ) methods like QuIP#[37], AQLM[11], and VPTQ [28] enable extremely low-bit quantization of model weights. However, these approaches do not mitigate the growing memory consumption of the KV cache, which scales linearly with sequence length during long-context inference.

To address this, recent studies have explored compressing the KV cache. KVQuant [19] and KIVI [30] observe the presence of outlier patterns in KV representations and propose to apply channel-wise quantization for key and token-wise quantization for value. CQ [43] extends non-uniform quantization (nuq) to multiple channels, achieving effective 1-bit KV cache compression. Orthogonal approaches, such as H2O [44] and SnapKV [25], reduce memory through cache eviction of uninformative tokens. Hybrid strategies like ZipCache [17] and MiKV [41] selectively maintain high-precision cache for important tokens while applying low-bit quantization elsewhere. Our method uses the same precision for all tokens, but future work could incorporate token importance as in hybrid approaches.

## 6  Conclusion

In this work, we propose NSNQuant, a calibration-free vector quantization (VQ) method for compressing KV cache of LLMs. NSNQuant effectively aligns the token distribution with the standard normal distribution through the NSN (Normalize-Shift-Normalize) process, enabling the use of the specialized codebook for the standard normal distribution. Through extensive experiments, we demonstrate that unlike calibration-based VQ methods, NSNQuant generalizes well across different tasks and datasets. In particular, NSNQuant excels in 1-bit quantization, outperforming the previous state-of-the-art method by a huge gap. We also implement efficient CUDA kernels for NSNQuant, and verify that NSNQuant achieves a $3\times$ speedup compared to the FP16 baseline.

## Acknowledgements

This work was supported by SK Hynix, Samsung Advanced Institute of Technology (SAIT), IITP/NRF grants funded by the Korean government (MSIT, 2021-0-00105 Development of Model Compression Framework for Scalable On-Device AI Computing on Edge Applications, 2021-0-01343 Artificial Intelligence Graduate School Program (Seoul National University), 2021-0-02068 Artificial Intelligence Innovation Hub), and Inter-university Semiconductor Research Center (ISRC), SNU. We are also grateful to VESSL AI for providing the compute resources used for the experiments in this work.

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

# A    Proof for variance bounds of NSN

## A.1    Proof of Lemma 1

**1. Notation.** Write $\mu_i = \mathbb{E}[X_i]$ and $\bar{\varepsilon} = \frac{1}{d}\sum_{i=1}^{d}\mu_i^2 \leq \varepsilon$. Decompose $\Sigma = \mathrm{Cov}(X) = D + A$, $D = \mathrm{diag}(\Sigma)$, $A_{ii} = 0$, $\|A\|_F \leq \Gamma$.

**2. Expected diagonal term.** Because $\sum_i \mathbb{E}[X_i^2] = d$,

$$\frac{1}{d}\,\mathrm{Tr}(D) = \frac{1}{d}\sum_{i=1}^{d}\big(\mathrm{Var}(X_i)\big) = 1 - \bar{\varepsilon}.$$

**3. Quadratic form.** Choose a Hadamard row $h = \frac{1}{\sqrt{d}}s$, $s \in \{\pm 1\}^d$. Then

$$h^{\mathsf{T}}Dh = 1 - \bar{\varepsilon}, \qquad f(h) := h^{\mathsf{T}}Ah = \frac{1}{d}s^{\mathsf{T}}As,$$

and $\mathrm{Var}(Y_i) = (1 - \bar{\varepsilon}) + f(h)$. $\mathbb{E}[f(h)] = 0$ since $s$ in randomized with the equal probability of $1/2$.

**4. Hanson–Wright** Since $s$ is a Rademacher vector, it is a sub-gaussian vector. Applying the Hanson-Wright inequality [34], for any $u > 0$

$$\Pr\big(|s^{\mathsf{T}}As| > u\big) \leq 2\exp\Big(-cu^2/\Gamma^2\Big)$$

where $c$ is a universal constant. Put $u = dt$; then

$$\Pr\big(|f(h)| > t\big) \leq 2\exp\Big(-c\,d^2t^2/\Gamma^2\Big).$$

**5. Tail parameter.** Choose $t = \Gamma\beta_\alpha$, $\beta_\alpha = \frac{1}{d}\sqrt{\ln(2/\alpha)/c}$. Exponent becomes $-\ln(2/\alpha)$; hence $\Pr(|f(h)| > t) \leq \alpha$.

**6. Combine.** Since $0 \leq \bar{\varepsilon} \leq \varepsilon$, with probability at least $1 - \alpha$

$$\mathrm{Var}(Y_i) \in \big[\,1 - \varepsilon - \Gamma\beta_\alpha,\ 1 + \Gamma\beta_\alpha\big].$$

## A.2    Off-diagonal Frobenius norms of covariance

To obtain insights regarding the bounds from Lemma 1, we measure the layer-wise average off-diagonal Frobenius norms of covariance matrixs. For key, the covariance is computed right before the Hadamard transform. For value, we remove fused Hadamard transform in the value projection matrix and compute covariance right after the NSN. Note again that the order between adjacent NSN and Hadamard transform is interchangable. The result is shown in Figure 6. The Frobenius norm is large in the first few layers for both key and value, and remains low in the later layers. This finding is consistent with the pattern observed in Figure 12 and 13, where standardization fails in the early layers. The reason for the large Frobenius norm in the first layers is visualized in Figure 7. For both key and value, there exist outlier values in covariance matrices, leading to the large Frobenius norm.

# B    Attention computation in NSN

NSNQuant computes the attention weights and outputs using the byproducts from the NSN process. First, the dot product between the query and key is computed as follows:

$$\begin{aligned}
qK^T &= q(\mathrm{RoPE}(K_{\text{pre-RoPE}}))^T = q(\mathrm{RoPE}\,(s_1^k(s_2^k v_{\text{nsn}}^k + o^k)))^T \\
&= q\,(s_1^k s_2^k\,\mathrm{RoPE}(v_{\text{nsn}}^k) + s_1^k\,\mathrm{RoPE}(o^k))^T \\
&= s_1^k s_2^k\,\mathrm{HT}(q)(\mathrm{HT}(\mathrm{RoPE}(v_{\text{nsn}}^k)))^T + s_1^k q(\mathrm{RoPE}(o^k))^T \\
&\simeq s_1^k s_2^k q_{\text{Had}} v_Q^{k\,T} + s_1^k q(\mathrm{RoPE}(o^k))^T
\end{aligned}$$

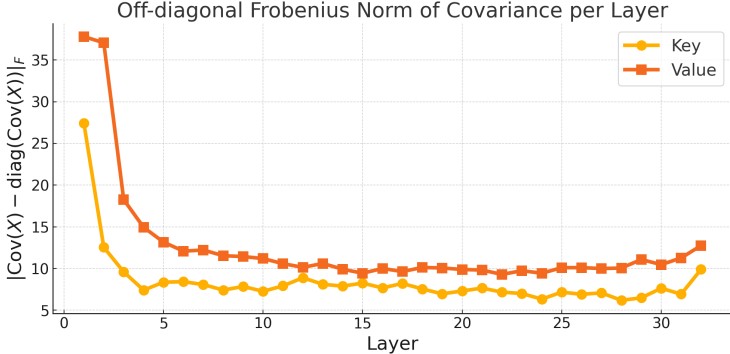

Figure 6: Average off-diagonal Frobenius norm of covariance matrix. The results are measured with LLaMA2-7B on WikiText-2.

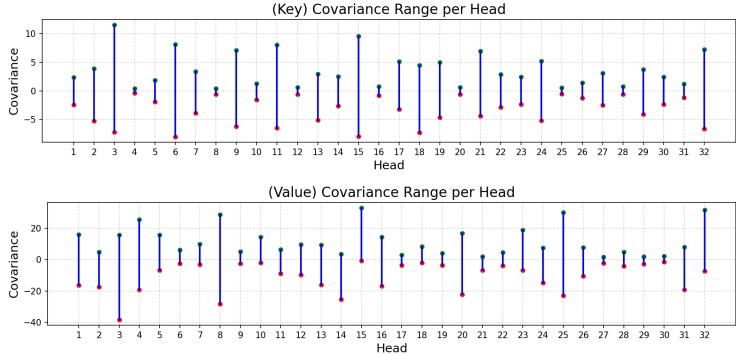

Figure 7: The min-max range of off-diagonal covariances of keys (top) and values (bottom) in the first layer of LLaMA2-7B. For both key and value, some channels suffer from outliers in covariance matrices. We use the first sample from the WikiText-2 for visualization.

Here, $\mathrm{HT}(\cdot)$ is the Hadamard transform, and $v_Q^k := \mathbb{C}[\mathrm{VQ}(\mathrm{HT}(\mathrm{RoPE}(v_{\mathrm{nsn}}^k)))]$, $q_{\mathrm{Had}} := \mathrm{HT}(q)$. Since $o^k$ has a different length from $K$, we expand it within a kernel to match the shape. We obtain attention weights through $W := \mathrm{softmax}(qK^T)$ and then compute the outputs as follows:

$$Wv = Ws_1^v(s_2^v v_{\mathrm{nsn}}^v + o^v) \simeq Ws_1^v(s_2^v v_Q^v + o^v), \quad v_Q^v := \mathbb{C}[\mathrm{VQ}(v_{\mathrm{nsn}}^v)].$$

## C   Additional ablation studies

### C.1   Effects of using randomized Hadamard transform (RHT)

From Lemma 1, we find that adopting randomized Hadamard transform (RHT) after the NSN process gives theoretical bounds to the variance of each channel. However, since the covariances between channels tend to have uniform signs, we find that using the naive Hadamard transform works well enough. As shown in Table 7, both transforms give similar results. Since RHT needs more parameters and computations, we use the naive Hadamard transform in the final design.

Table 7: Perplexity of LLaMA2-7B on WikiText-2 and C4 with NSNQuant-2b using different types of Hadamard transform. We follow the settings from the main experiments for WikiText-2 and C4. For LCC and SAMSum, we truncate each data sample to 4096 tokens if needed.

| Method | WikiText-2 | C4 | LCC | SAMSum |
|---|---|---|---|---|
| Hadamard transform | **5.285** | 6.856 | **2.128** | 6.466 |
| RHT | 5.290 | **6.854** | **2.128** | **6.461** |

## C.2 Effects of residual size

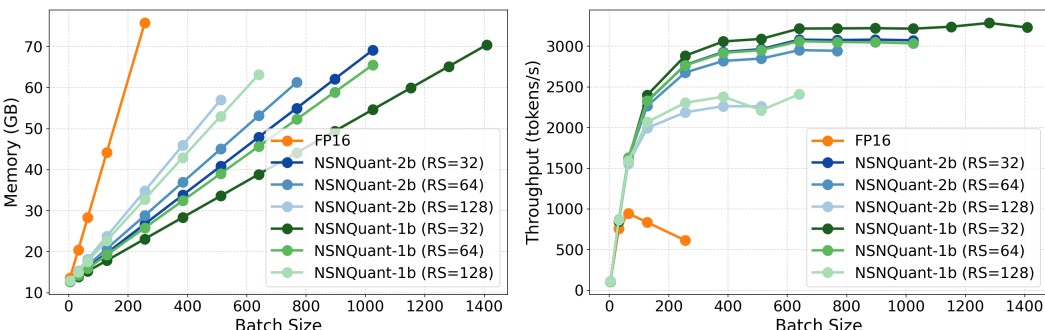

Figure 8: Peak memory usage (left) and throughput (right) measured with varying batch sizes and residual sizes

Table 8: Evaluation results on GSM8K, HumanEval, CoQA, and MMLU with LLaMA3.1-8B-Instruct. RS refers to residual size.

| Method | Bits | GSM8K (8-shot, CoT) | HumanEval | CoQA | MMLU (4-shot, CoT) | | | |
|---|---|---|---|---|---|---|---|---|
| | | | | | Humanities | STEM | Social | Other |
| NSNQuant-2b (RS=32) | 2.30 | 74.90 | 53.05 | **63.83** | 70.24 | 56.56 | 74.22 | **70.98** |
| NSNQuant-2b (RS=64) | 2.23 | **75.89** | 56.10 | **63.83** | **71.04** | 55.64 | 73.42 | 70.74 |
| NSNQuant-2b (RS=128) | 2.19 | 75.66 | **56.71** | 62.93 | 69.29 | **57.38** | **74.49** | 70.80 |
| NSNQuant-1b (RS=32) | 1.30 | 42.08 | 42.07 | 63.67 | 59.47 | 44.88 | 65.95 | 63.42 |
| NSNQuant-1b (RS=64) | 1.23 | 53.45 | 44.51 | 62.70 | 59.82 | 45.83 | 65.34 | 63.77 |
| NSNQuant-1b (RS=128) | 1.19 | **57.54** | **48.17** | **63.88** | **61.77** | **49.07** | **67.63** | **66.86** |

We fix the residual size to 64 in our main experiments, for fair evaluation across different methods. However, the residual size is an important hyperparameter that determines the number of full precision caches. Therefore, we evaluate NSNQuant with 3 different residual sizes (32, 64, 128) with LLaMA3.1-8B-Instruct. Since LongBench is not effective to show performance differences in 2-bit regime, we use GSM8K, HumanEval, and CoQA. The result is shown in Table 8. For NSNQuant-1b, we observe that performance is highly sensitive to the choice of residual size. A residual size of 128 yields the best results across most benchmarks, while a residual size of 32 performs the worst. In contrast, NSNQuant-2b shows comparable performance with different residual sizes. These results suggest that NSNQuant-2b produces higher-quality quantizations, making it more robust to variations in residual size.

We also report the memory usage and throughput with different residual sizes in Figure 8. The result shows that the methods with smaller residual sizes require less memory and achieve higher throughput. Note that the impact of residual size on memory and throughput will decrease as the sequence length gets longer.

## C.3 Effects of scale adjustment

**Motivating Example** Suppose we are quantizing a 2-dimensional vector $(1, 2)$ using a codebook with two entries: $(0.8, 1.6)$ and $(2, 3)$. The vector would be approximated by $(0.8, 1.6)$, incurring a non-negligible error. However, this error could be significantly reduced by scaling the quantized vector by a factor of $1.25$.

As this example illustrates, using the quantized vector $v_Q$ without any scaling leads to suboptimal approximation. To identify a more effective scaling strategy, we evaluate three probable approaches, as visualized in Figure 9. The first approach is to scale $v_Q$ to minimize the L2 error between $v$ and $v_Q$, the second approach is to scale $v_Q$ to match the size of $v$, and the third approach is to scale $v_Q$ to preserve components parallel to $v$. The result of applying three strategies is presented in Table 9. For both key and value, strategy 3 shows the best quantization quality. By applying it to both key and

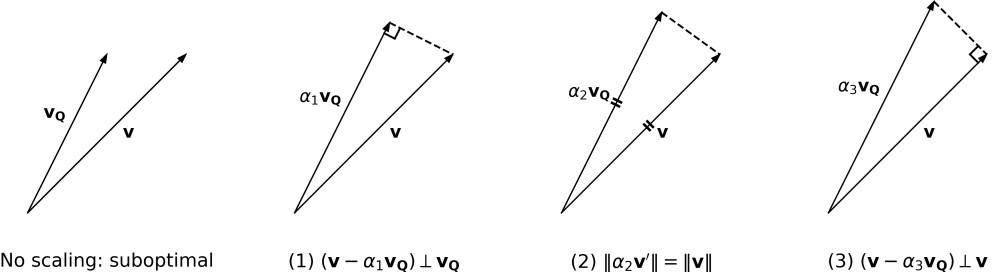

Figure 9: Visualization of the three tested scale adjustment strategies.

Table 9: Perplexity of LLaMA2-7B on WikiText-2 under different scaling strategies for key and value. NSNQuant-2b is used here for ablation.

| Strategy | PPL |
|---|---|
| No scaling | 5.395 |
| Key - Strategy 1 | 5.378 |
| Key - Strategy 2 | 5.329 |
| Key - Strategy 3 | **5.317** |
| Value - Strategy 1 | 5.394 |
| Value - Strategy 2 | 5.355 |
| Value - Strategy 3 | **5.335** |
| Key - Strategy 3 + Value - Strategy 3 | **5.285** |

value in NSNQuant-2b, the perplexity of LLaMA2-7B on WikiText-2 is reduced from 5.395 to 5.285.

## C.4 Effects of codebook tuning

Table 10: Perplexity of LLaMA2-7B on WikiText-2 using different codebooks.

| Method | Codebook | PPL |
|---|---|---|
| | K-Means | 5.294 |
| NSNQuant-2b | E8P | **5.284** |
| | K-Means + Fine-tuning | 5.285 |
| NSNQuant-1b | K-Means | 6.910 |
| | K-Means + Fine-tuning | **6.703** |

As mentioned in section 3.4, we propose to fine-tune a codebook obtained from the K-Means algorithm. Since the error of scale adjustment only depends on the angle between $v$ and $v_Q$, we set the objective to minimize the cosine distance between them. The perplexity result with LLaMA2-7B on WikiText-2 is presented in Table 10. The result shows that fine-tuning improves quantization quality in both NSNQuant-1b and NSNQuant-2b, and matches the performance of E8P [37] in 2-bit quantization.

## C.5 Effects of pre-RoPE NSN

As visualized in Figure 2, NSNQuant applies NSN to keys before the RoPE. However, it seems natural to apply NSN after RoPE since RoPE may induce noises to the channel-wise mean. The PPL evaluation results in both settings are presented in Table 11. The pre-RoPE NSN achieves slightly lower PPL than the post-RoPE NSN. We attribute this to two reasons. First, the effect of RoPE on the channel-wise mean is minimal, as the tokens within the residual share similar rotation angles. Second, computing full-precision RoPE for $o$ is beneficial, since the RoPE rotation matrix contains important position information.

Table 11: Perplexity of LLaMA2-7B and LLaMA2-13B on WikiText-2 with NSNQuant-2b using different locations of NSN.

| Normalization | LLaMA2-7B | LLaMA2-13B |
|---|---|---|
| pre-RoPE NSN | **5.29** | **4.71** |
| post-RoPE NSN | 5.33 | 4.73 |

## C.6 Effects of double quantization

Table 12 shows the performance change when applying double quantization. The performance degradation is minimal, while the memory savings are substantial.

Table 12: Impact of double quantization on the perplexity of LLaMA2-7B on WikiText-2, using NSNQuant-2b for ablation.

| Method | Avg. bit width | PPL |
|---|---|---|
| Baseline | 2.5 | 5.278 |
| + Codebook Double Quantization | 2.5 | 5.280 |
| + Mean Double Quantization | 2.33 | 5.287 |
| + Scale Double Quantization | 2.23 | 5.285 |

## C.7 Scaling calibration set for CQ

Since CQ uses only a very small dataset of 16 samples, a natural question arises: **can CQ be further improved to outperform NSNQuant by scaling its calibration set?** To answer this question, we test 2 variants of CQ where the size of the calibration set is doubled to 32. For the first variant, we use 32 samples from WikiText-2 (W32), enlarging the size of the calibration set. For the other variant, we use 16 samples from WikiText-2 and another 16 samples from C4 (W16C16). This variant not only increases the size but diversifies the calibration set, covering a wider range of input distribution. We use LLaMA3-8B-Instruct model to test performance on the three downstream tasks: HumanEval, GSM8K, and CoQA. We denote two variants as W32 and W16C16, respectively.

The result is shown in Table 13. While W32 shows only marginal improvement from the baseline, W16C16 provides clearer benefits. It suggests that to improve the robustness of CQ, it is important to diversify the calibration set, rather than just scaling it up. In addition, despite improved performance, NSNQuant still shows better results. For example, on HumanEval, CQ-8c10b still suffers from significant performance degradation since its input distribution (i.e. codes) is not covered by either WikiText-2 or C4.

We believe that further scaling of the calibration set of CQ will improve its generalization ability. However, since CQ needs gradient computation, the calibration process incurs comparable cost to fine-tuning, as the calibration set grows. Moreover, it's challenging to prepare the calibration set that adequately covers the full diversity of real-world input distributions (e.g. different languages). Therefore, we believe that NSNQuant remains a highly practical and robust option in general scenarios, especially when broad generalization is required

Table 13: Performance comparison of NSNQuant and CQ baselines on HumanEval, GSM8K (8-shot, CoT), and CoQA.

| Method | HumanEval | GSM8K (8-shot, CoT) | CoQA |
|---|---|---|---|
| FP16 | 28.05 | 76.62 | 61.47 |
| **NSNQuant-2b** | **32.93** | **74.53** | **61.62** |
| CQ-4c9b | 23.78 | 72.55 | 60.60 |
| CQ-4c9b (W32) | 25.00 | 72.93 | 60.38 |
| CQ-4c9b (W16C16) | 29.27 | 73.16 | 60.98 |
| **NSNQuant-1b** | **29.27** | **61.64** | **60.28** |
| CQ-8c10b | 13.41 | 24.26 | 53.38 |
| CQ-8c10b (W32) | 15.24 | 21.76 | 53.37 |
| CQ-8c10b (W16C16) | 12.20 | 46.47 | 54.07 |

# D  Variants of NSN

## D.1  Replacing the Shift step with Weiszfeld algorithm

As noted in Section 3.2, the final normalization step of NSN can introduce a small bias, leaving the channel-wise mean slightly off zero. To verify that it barely affects the quantization quality, we provide a mathematical explanation and an experiment where we totally remove such bias by adopting Weiszfeld algorithm.

First, here is an intuitive explanation of why the small bias introduced by the last step of NSN does not significantly affect the quantization error. Let $v_{\mathrm{n}} = \{v_1, v_2, ..., v_l\} \in \mathbb{R}^{l \times d}$ denote the output tokens after the first normalization step. Since the next step, **Shift**, subtracts the mean, we have $v_{\mathrm{ns}} = \{v_1 - \mathbb{E}[v_i], v_2 - \mathbb{E}[v_i], ..., v_l - \mathbb{E}[v_i]\}$. **(i)** If $\mathrm{Var}[v_i]$ is small, then $\mathbb{E}[(v_i - \mathbb{E}[v_i])^2]$ is also small. Since we save $o = \mathbb{E}[v_i]$ in full-precision, the fact that the norm of the leftover part $(v_i - \mathbb{E}[v_i])$ is small implies that the final error will also be small. **(ii)** If $\mathrm{Var}[v_i]$ is large, then $\mathbb{E}[v_i]$ will be small because $\mathbb{E}[v_i]^2 = \mathbb{E}[v_i^2] - \mathrm{Var}[v_i] = d - \mathrm{Var}[v_i]$. Small $\mathbb{E}[v_i]$ means that the token-wise norm will not change much, and the last step will not significantly affect the channel-wise mean. To conclude, regardless of the magnitude of $\mathrm{Var}[v_i]$, the final quantization error would remain small because either the alignment is well preserved, or the leftover magnitude is small.

Second, to empirically explore the impact of removing this drift, we reformulate the second step (Shift) as the search for the geometric median. Let $\{t_i\}_{i=1}^{\mathrm{rs}} \subset \mathbb{R}^d$ denote the tokens in a group of residual size after the first normalization step. We seek $o_\star$ such that

$$\mathcal{L}(o) := \sum_{i=1}^{\mathrm{rs}} \frac{t_i - o}{\|t_i - o\|} = \mathbf{0}. \tag{2}$$

Setting $F(o) := \sum_{i=1}^{\mathrm{rs}} \|t_i - o\|$ gives $\nabla F(o) = \mathcal{L}(o)$; hence any $o$ satisfying (2) minimizes $F$ and is the geometric median of the set $\{t_i\}$.

We compute $o_\star$ with the Weiszfeld algorithm—the standard iterative solver for geometric medians—and subtract it instead of the arithmetic mean in NSN's second step. The resulting variant keeps the post-normalization mean essentially at zero. Table 14 presents the perplexity evaluation results on WikiText-2 and C4 with LLaMA2-7B. The change yields only marginal negative effects despite the additional cost from the iterative updates. Hence, we keep the original mean-subtraction step in the final NSN design.

Table 14: Perplexity of LLaMA2-7B on WikiText-2 and C4 with NSNQuant-2b using different normalization strategies.

| Normalization | WikiText-2 | C4 |
|:---:|:---:|:---:|
| NSN | **5.285** | **6.856** |
| N-Weiszfeld-N | 5.289 | 6.863 |

## D.2  Replacing the second Normalization step with channel-wise scaling

A straightforward way to standardize the output distribution is to simply shift and scale in a channel dimension. Therefore, we try replacing our third step—token-wise normalization—with channel-wise scaling, where sample standard deviation is divided from each channel. We also move the location of normalizations next to the Hadamard transform so that it does not affect channel-wise statistics. The result is shown in Table 15. Using channel-wise scaling instead of token-wise normalization is not as effective as token-wise normalization. This is because channel-wise scaling does not suppress outlier tokens, especially the attention sink token. For example, using the first sample from WikiText-2 dataset, the average norm of the first token in the first layer is 19.1, which is nearly twice as large as the overall average (11.3). Considering a ball-shaped property of our codebook, quantization error would be large for these tokens since their magnitudes are far from zero. On the other hand, our token-wise normalization effectively regulates the scale, making our codebook work effectively.

Table 15: Perplexity of LLaMA2-7B on WikiText-2 and C4 with NSNQuant-2b using different normalization strategies

| Normalization | WikiText-2 | C4 |
|---|---|---|
| NSN | **5.285** | **6.856** |
| NS-Channel-wise Scaling | 6.251 | 8.265 |

# E  Experiment details

## E.1  Experiment environments

The PPL evaluation is performed on a Linux server with 2 RTX Titan GPUs. The evaluations on LongBench, GSM8K, HumanEval, CoQA, MMLU are conducted on a Linux server with 8 RTX 3090 GPUs. Efficiency analysis is conducted on a Linux server with a single A100-80GB GPU. All methods are implemented based on the HuggingFace Transformers library using PyTorch framework.

## E.2  LongBench evaluation metrics

Table 16 shows the evaluation metrics used in LongBench. Qasper, TREC, and TriviaQA use exact matching-based metrics, while the other tasks use heuristic metrics.

Table 16: Evaluation metrics used in LongBench evaluation

| Task | Evaluation Metric |
|---|---|
| Qasper | F1 |
| QMSum | ROUGE-L |
| MultiNews | ROUGE-L |
| TREC | Accuracy |
| TriviaQA | F1 |
| SAMSum | ROUGE-L |
| LCC | Edit Sim |
| RepoBench-P | Edit Sim |

## E.3  Additional explanation on baselines

**KIVI** [30] is one of the pioneering works in low-bit quantization of the KV cache of LLMs. KIVI quantizes key cache along the channel dimension and value cache along the token dimension, considering its outlier patterns. To maintain consistency with our residual policy, we use group size of 64 for keys. Also, since smaller group sizes incur large additional bits which is critical for low-bit scenario, we use 128 for values.

**KIVI + Had** is a variant of KIVI, where the Hadamard transform for key and value is added to the existing KIVI framework. We add this variant because Hadamard transform is known to reduce errors in token-wise quantization a lot, especially in a round-to-nearest (RTN)-based method like KIVI.

**KVQuant** [19] is another pioneering work in KV cache quantization. Similar to KIVI, KVQuant quantizes key and value cache along the channel dimension and token dimension, respectively. KVQuant employs non-uniform quantization (nuq) and dense-and-sparse quantization to improve performance further. We apply Q-Norm and 1% dense-and-sparse quantization to both KVQuant-1b and KVQuant-2b to obtain the best result.

**CQ** [43] is our main competitor, which shares the core idea of VQ. CQ quantizes grouped channels using learned centroids, which serve as a codebook. We evaluate the performance of CQ-4c9b for 2-bit quantization, and CQ-8c10b for 1-bit quantization.

### E.4 LM-Eval tasks

For evaluation on GSM8K, HumanEval, CoQA, and MMLU, we use the following tasks provided in `lm-evaluation-harness` [13]: `gsm8k_cot`, `humaneval`, `coqa`, `mmlu_flan_cot_fewshot_humanities`, `mmlu_flan_cot_fewshot_stem`, `mmlu_flan_cot_fewshot_social_sciences`, `mmlu_flan_cot_fewshot_other`.

# F   Additional results

## F.1   PPL evaluation in generation setting

Since the perplexity (PPL) evaluation in Table 2 is different from our generation setting, we provide PPL evaluation results in the generation setting. We obtain the output logits by processing tokens one-by-one, just like in a generation scenario. We also adopt residuals to maintain recent tokens in full-precision. The result is presented in Table 17. All methods achieve lower PPLs compared to the results in Table 2 because of the residual, but their relative order exhibits similar trends. Since the PPL evaluation setting in the main table is much more efficient to measure and closer to convention in the previous studies, we use the same setting in the ablation studies.

Table 17: Perplexity measured in the generation setting with residual. We evaluate LLaMA2-7B and LLaMA3.1-8B on WikiText-2 and C4.

| Method | Avg. bit width | Dataset | LLaMA2-7B | LLaMA3.1-8B |
|---|---|---|---|---|
| FP16 | 16 | C4 | 7.04 | 10.36 |
| | | WikiText-2 | 5.42 | 7.04 |
| KIVI-2 | 2.38 | C4 | 6.63 | 8.43 |
| | | WikiText-2 | 5.12 | 5.84 |
| KIVI-2 + Had | 2.38 | C4 | 6.88 | 9.56 |
| | | WikiText-2 | 5.30 | 6.68 |
| KVQuant-2b + 1% | 2.32 | C4 | 6.84 | 8.88 |
| | | WikiText-2 | 5.32 | 6.20 |
| CQ-4c9b | 2.26 | C4 | 6.84 | 9.48 |
| | | WikiText-2 | 5.22 | **6.00** |
| NSNQuant-2b | 2.23 | C4 | **6.69** | **8.69** |
| | | WikiText-2 | **5.16** | 6.03 |
| KVQuant-1b + 1% | 1.32 | C4 | 9.67 | 16.05 |
| | | WikiText-2 | 6.70 | 12.02 |
| CQ-8c10b | 1.27 | C4 | 7.61 | 20.33 |
| | | WikiText-2 | 5.56 | **6.53** |
| NSNQuant-1b | 1.23 | C4 | **7.07** | **10.32** |
| | | WikiText-2 | **5.48** | 7.55 |

## F.2   Additional results on LongBench

Table 18 presents evaluation results on LongBench with LLaMA2-7B-Chat and LLaMA3-8B-Instruct. Similar to the main table, NSNQuant-1b outperforms other 1-bit quantization methods by a large margin. However, 2-bit results are quite noisy, without any clear trend. For example, KVQuant-2b struggles in LLaMA2-7B-Chat, but achieves the best score in LLaMA3-8B-Instruct. This is another example that shows the noisiness of metrics. As shown in Table 18, KVQuant-2b achieves 56.51 in RepoBench-P, which is much higher than that of FP16. This result is unreliable since KVQuant suffers from performance degradation in other tasks.

To reduce the impact of noisiness, we evaluate each method from a different perspective: how well each method preserves the original outputs. To achieve this, we measure the ROUGE-L score of each method by comparing their outputs with FP16 outputs. Since Qasper, TREC, and TriviaQA are evaluated using exact matching-based metrics, we exclude them from the task list. The result is shown in Table 19. It clearly shows that NSNQuant is the most effective method which preserves the original output faithfully. On the other hand, CQ is even worse than KIVI + Had, suggesting that calibration-based VQ suffers from significant performance degradation when applied to diverse tasks.

Table 18: Additional results on LongBench with LLaMA2-13B-Chat, LLaMA2-7B-Chat and LLaMA3-8B-Instruct

| Model | Method | Bits | Qasper | QMSum | MultiNews | TREC | TriviaQA | SAMSum | LCC | RepoBench-P | Avg. |
|---|---|---|---|---|---|---|---|---|---|---|---|
| LLaMA2-13B-Chat | FP16 | 16 | 17.06 | 20.95 | 26.55 | 68.50 | 87.75 | 42.59 | 48.27 | 49.80 | 45.18 |
| | KIVI-2 | 2.38 | 17.44 | 20.53 | 26.03 | 67.00 | 87.39 | 41.79 | 46.54 | 47.33 | 44.26 |
| | KIVI-2 + Had | 2.38 | 15.44 | 19.59 | 26.19 | 68.00 | 86.55 | 41.93 | 48.22 | 50.10 | 44.50 |
| | KVQuant-2b + 1% | 2.32 | 16.57 | 19.72 | 25.59 | 68.00 | 88.07 | 40.72 | 47.64 | 49.70 | 44.50 |
| | CQ-4c9b | 2.26 | 18.42 | 19.72 | 25.12 | 67.00 | 87.69 | 41.32 | 47.18 | 48.32 | 44.35 |
| | NSNQuant-2b | 2.23 | 17.28 | 20.41 | 26.16 | 68.50 | 87.51 | 42.48 | 47.82 | 49.69 | **44.98** |
| | KVQuant-1b + 1% | 1.32 | 13.85 | 18.32 | 20.11 | 46.50 | 81.67 | 29.60 | 35.46 | 32.78 | 34.79 |
| | CQ-8c10b | 1.27 | 16.08 | 18.67 | 20.87 | 49.00 | 87.12 | 37.44 | 43.17 | 42.34 | 39.34 |
| | NSNQuant-1b | 1.23 | 17.98 | 20.56 | 25.92 | 67.50 | 87.17 | 41.08 | 48.19 | 50.10 | **44.81** |
| LLaMA2-7B-Chat | FP16 | 16 | 21.95 | 20.71 | 26.21 | 64.00 | 83.09 | 41.39 | 58.31 | 52.16 | 45.98 |
| | KIVI-2 | 2.38 | 24.69 | 20.83 | 25.99 | 63.50 | 83.05 | 40.57 | 56.69 | 49.90 | 45.65 |
| | KIVI-2 + Had | 2.38 | 20.62 | 21.03 | 25.98 | 64.00 | 83.45 | 41.11 | 56.79 | 51.10 | 45.51 |
| | KVQuant-2b + 1% | 2.33 | 20.84 | 21.22 | 24.89 | 62.50 | 83.82 | 39.87 | 55.48 | 49.73 | 44.79 |
| | CQ-4c9b | 2.26 | 20.73 | 20.64 | 24.92 | 63.00 | 84.14 | 39.90 | 57.63 | 51.30 | 45.28 |
| | NSNQuant-2b | 2.23 | 22.01 | 20.87 | 26.42 | 64.00 | 83.67 | 40.51 | 57.71 | 51.47 | **45.83** |
| | KVQuant-1b + 1% | 1.32 | 13.10 | 20.27 | 20.82 | 30.00 | 61.43 | 34.14 | 43.16 | 39.31 | 32.78 |
| | CQ-8c10b | 1.27 | 14.82 | 19.82 | 20.48 | 49.00 | 82.49 | 36.62 | 49.45 | 46.60 | 39.04 |
| | NSNQuant-1b | 1.23 | 17.70 | 20.74 | 25.49 | 64.00 | 82.06 | 40.31 | 55.14 | 50.57 | **44.50** |
| LLaMA3-8B-Instruct | FP16 | 16 | 31.25 | 23.54 | 26.69 | 74.00 | 90.31 | 42.65 | 57.23 | 51.69 | 49.67 |
| | KIVI-2 | 2.38 | 20.92 | 23.82 | 26.34 | 74.00 | 90.08 | 40.87 | 46.64 | 46.87 | 46.19 |
| | KIVI-2 + Had | 2.38 | 23.93 | 22.58 | 26.35 | 74.00 | 90.01 | 41.38 | 49.97 | 47.11 | 46.92 |
| | KVQuant-2b + 1% | 2.32 | 27.91 | 23.16 | 25.26 | 74.00 | 90.33 | 39.95 | 56.50 | 56.51 | **49.20** |
| | CQ-4c9b | 2.26 | 26.26 | 22.60 | 25.10 | 73.50 | 90.78 | 40.48 | 57.89 | 53.28 | 48.74 |
| | NSNQuant-2b | 2.23 | 29.83 | 22.66 | 26.28 | 74.00 | 90.31 | 41.73 | 55.88 | 50.31 | 48.88 |
| | KVQuant-1b + 1% | 1.32 | 13.01 | 21.45 | 21.39 | 61.50 | 86.12 | 34.23 | 47.89 | 45.96 | 41.44 |
| | CQ-8c10b | 1.27 | 15.23 | 21.04 | 21.63 | 44.50 | 87.26 | 36.63 | 51.24 | 46.36 | 40.49 |
| | NSNQuant-1b | 1.23 | 18.04 | 21.57 | 26.36 | 73.50 | 90.46 | 41.06 | 45.50 | 42.19 | **44.84** |

### F.3 Additional results on GSM8K, HumanEval, CoQA, and MMLU

Table 20 presents the additional results for GSM8K, HumanEval, CoQA, and MMLU. The result shows a similar trend to Table 4, where NSNQuant generally outperforms other methods in both 1-bit and 2-bit quantization.

### F.4 Additional latency breakdown results

Table 21 presents additional latency breakdown results under different configurations, measured with LLaMA2-7B. We fix the number of generated tokens to 64 and vary the batch size and input prompt length to observe performance trends. The results exhibit a similar pattern to Table 5, where NSNQuant incurs additional overhead during the prefill stage but demonstrates lower latency during the decode stage.

### F.5 Results on AIME-2024

To evaluate the long-context reasoning ability, we evaluate our methods on AIME-2024, with DeepSeek-R1-Distill-Llama-8B. As suggested in DeepSeek [15], we set temperature to 0.6, top-p to 0.95, and maximum generated tokens to 32768. Since AIME-2024 only contains 30 problems, we run the evaluation with 5 different seeds.

The result is shown in Table 22. In 2-bit quantization, NSNQuant and CQ both show little accuracy drop with small differences. On the other hand, under 1-bit quantization, both methods experience significant performance degradation, but NSNQuant-1b achieves over $2\times$ higher accuracy than CQ-8c10b. While CQ-4c9b is generally strong in math reasoning tasks such as GSM8K and AIME-2024, CQ-8c10b fails severely in such datasets.

### F.6 Comparison with QuaRot and SpinQuant

QuaRot [2] and SpinQuant [29] are the recent quantization methods for KV cache which also leverage the Hadamard transform. However, we don't include them as our baselines because they

Table 19: Average ROUGE-L score measured with FP16 output. Qasper, TREC and TriviaQA are excluded since these tasks provide the objective metrics based on exact matching.

| Model | Method | Bits | QMSum | MultiNews | SAMSum | LCC | RepoBench-P | Avg. |
|---|---|---|---|---|---|---|---|---|
| LLaMA2-13B-Chat | KIVI | 2.38 | 52.34 | 52.18 | 73.17 | 52.51 | 49.06 | 55.85 |
| | KIVI + Had | 2.38 | 55.24 | 55.39 | 76.53 | 60.55 | 57.24 | _60.99_ |
| | KVQuant-2b + 1% | 2.32 | 55.85 | 53.02 | 76.55 | 57.07 | 53.91 | 59.28 |
| | CQ-4c9b | 2.26 | 56.07 | 50.77 | 74.61 | 54.85 | 49.19 | 57.10 |
| | NSNQuant-2b | 2.23 | 62.73 | 59.69 | 85.02 | 70.93 | 68.23 | **69.32** |
| | KVQuant-1b + 1% | 1.32 | 40.88 | 32.00 | 42.86 | 22.95 | 18.32 | 31.40 |
| | CQ-8c10b | 1.27 | 43.39 | 34.49 | 59.21 | 30.59 | 26.41 | _38.82_ |
| | NSNQuant-1b | 1.23 | 51.63 | 51.12 | 71.77 | 52.94 | 48.44 | **55.18** |
| LLaMA2-7B-Chat | KIVI | 2.38 | 46.95 | 48.04 | 67.22 | 47.32 | 46.96 | 51.30 |
| | KIVI + Had | 2.38 | 49.12 | 50.13 | 73.87 | 55.30 | 56.36 | _56.95_ |
| | KVQuant-2b + 1% | 2.32 | 50.77 | 47.71 | 74.73 | 55.20 | 55.24 | 56.73 |
| | CQ-4c9b | 2.26 | 52.04 | 47.19 | 69.22 | 50.70 | 48.80 | 53.59 |
| | NSNQuant-2b | 2.23 | 58.19 | 55.42 | 80.14 | 69.93 | 67.36 | **66.21** |
| | KVQuant-1b + 1% | 1.32 | 37.66 | 33.14 | 44.16 | 22.34 | 23.54 | 32.17 |
| | CQ-8c10b | 1.27 | 40.01 | 31.07 | 56.00 | 25.52 | 27.02 | _35.92_ |
| | NSNQuant-1b | 1.23 | 46.42 | 47.91 | 66.08 | 47.65 | 46.92 | **51.00** |
| LLaMA3-8B-Instruct | KIVI | 2.38 | 49.38 | 48.54 | 44.29 | 44.45 | 40.33 | 45.40 |
| | KIVI + Had | 2.38 | 49.35 | 51.77 | 46.83 | 49.65 | 46.67 | 48.85 |
| | KVQuant-2b + 1% | 2.32 | 50.60 | 50.00 | 45.66 | 53.18 | 46.14 | _49.12_ |
| | CQ-4c9b | 2.26 | 50.06 | 47.76 | 49.25 | 49.65 | 41.63 | 47.67 |
| | NSNQuant-2b | 2.23 | 57.24 | 57.02 | 55.29 | 65.43 | 60.09 | **59.01** |
| | KVQuant-1b + 1% | 1.32 | 37.22 | 31.72 | 29.58 | 25.71 | 22.70 | 29.39 |
| | CQ-8c10b | 1.27 | 37.92 | 32.27 | 34.20 | 28.07 | 22.25 | _30.94_ |
| | NSNQuant-1b | 1.23 | 44.00 | 48.31 | 41.04 | 42.99 | 36.17 | **42.50** |
| LLaMA3.1-8B-Instruct | KIVI | 2.38 | 45.23 | 47.48 | 42.19 | 52.31 | 43.98 | 46.24 |
| | KIVI + Had | 2.38 | 47.75 | 50.65 | 44.11 | 57.71 | 51.76 | _50.40_ |
| | KVQuant-2b + 1% | 2.32 | 46.33 | 49.32 | 45.85 | 57.20 | 52.07 | 50.15 |
| | CQ-4c9b | 2.26 | 45.45 | 48.54 | 47.76 | 54.13 | 45.55 | 48.29 |
| | NSNQuant-2b | 2.23 | 51.72 | 56.69 | 52.27 | 68.32 | 64.37 | **58.67** |
| | KVQuant-1b + 1% | 1.32 | 34.62 | 33.02 | 32.44 | 26.67 | 22.68 | _29.89_ |
| | CQ-8c10b | 1.27 | 33.70 | 32.47 | 29.32 | 29.36 | 22.30 | 29.43 |
| | NSNQuant-1b | 1.23 | 41.92 | 46.91 | 40.68 | 46.85 | 37.56 | **42.78** |
| Mistral-7B-Instruct-v0.3 | KIVI | 2.38 | 50.56 | 49.99 | 78.84 | 61.02 | 49.35 | 57.95 |
| | KIVI + Had | 2.38 | 56.40 | 55.62 | 79.54 | 65.26 | 57.83 | _62.93_ |
| | KVQuant-2b + 1% | 2.32 | 53.09 | 51.12 | 77.38 | 64.04 | 56.96 | 60.52 |
| | CQ-4c9b | 2.26 | 52.58 | 51.55 | 80.26 | 60.15 | 52.80 | 59.46 |
| | NSNQuant-2b | 2.23 | 63.19 | 60.06 | 84.42 | 75.32 | 69.85 | **70.57** |
| | KVQuant-1b + 1% | 1.32 | 36.65 | 32.30 | 60.32 | 34.81 | 28.97 | 38.61 |
| | CQ-8c10b | 1.27 | 39.17 | 33.26 | 64.46 | 34.86 | 27.29 | _39.81_ |
| | NSNQuant-1b | 1.23 | 49.96 | 50.01 | 74.29 | 55.03 | 45.03 | **54.86** |

Table 20: Additional results on GSM8K, HumanEval, CoQA, and MMLU with LLaMA2-13B-Chat and LLaMA3-8B-Instruct.

| Model | Method | Bits | GSM8K (8-shot, CoT) | HumanEval | CoQA | MMLU (4-shot, CoT) | | | |
|---|---|---|---|---|---|---|---|---|---|
| | | | | | | Humanities | STEM | Social | Other |
| LLaMA2-13B-Chat | FP16 | 16 | 37.30 | 17.07 | 64.08 | 60.89 | 41.54 | 62.54 | 60.95 |
| | KIVI | 2.38 | 30.63 | 15.24 | 63.70 | 57.48 | 38.66 | 57.42 | 56.21 |
| | KIVI + Had | 2.38 | 31.69 | 14.02 | 63.03 | **59.91** | _40.21_ | _61.05_ | _59.25_ |
| | KVQuant-2b + 1% | 2.32 | 32.98 | _15.85_ | **64.82** | 59.15 | 39.31 | 59.64 | 57.96 |
| | CQ-4c9b | 2.26 | _34.04_ | 14.63 | _64.60_ | 56.26 | 35.70 | 57.94 | 55.47 |
| | NSNQuant-2b | 2.23 | **35.48** | **18.29** | 63.67 | _59.16_ | **41.47** | **62.34** | **60.26** |
| | KVQuant-1b + 1% | 1.32 | 8.64 | 9.15 | 57.08 | _29.89_ | _23.14_ | _35.90_ | _21.44_ |
| | CQ-8c10b | 1.27 | _19.71_ | _12.20_ | _60.88_ | 21.95 | 11.25 | 30.84 | 19.33 |
| | NSNQuant-1b | 1.23 | **26.84** | **14.02** | **63.10** | **56.43** | **38.15** | **58.00** | **57.71** |
| LLaMA3-8B-Instruct | FP16 | 16 | 76.72 | 28.05 | 61.47 | 70.51 | 53.00 | 70.89 | 70.66 |
| | KIVI | 2.38 | 65.35 | 22.56 | 60.42 | 59.18 | 44.14 | 61.82 | 61.73 |
| | KIVI + Had | 2.38 | 69.75 | **32.93** | _60.72_ | 63.56 | 47.36 | 63.05 | 65.06 |
| | KVQuant-2b + 1% | 2.32 | 70.74 | 23.78 | 60.30 | _64.94_ | 46.34 | _64.67_ | _66.11_ |
| | CQ-4c9b | 2.26 | _72.55_ | 23.78 | 60.60 | 63.86 | _47.94_ | 64.20 | 65.26 |
| | NSNQuant-2b | 2.23 | **74.53** | **32.93** | **61.62** | **68.95** | **52.44** | **69.36** | **68.91** |
| | KVQuant-1b + 1% | 1.32 | 6.97 | _14.02_ | 51.28 | 11.21 | 13.73 | 31.25 | _26.97_ |
| | CQ-8c10b | 1.27 | _24.26_ | 13.41 | _53.38_ | _22.05_ | _14.44_ | _26.42_ | 16.14 |
| | NSNQuant-1b | 1.23 | **61.64** | **29.27** | **60.28** | **58.73** | **42.70** | **59.76** | **60.94** |

Table 21: Latency (ms) breakdown of NSNQuant-2b compared with the FP16 baseline. BS, L, HT and VQ refer to batch size, input prompt length, Hadamard transform and vector quantization, respectively. The number of generated tokens is fixed to 64.

| BS | L | Stage | Method | Total | NSN | HT | VQ |
|---|---|---|---|---|---|---|---|
| 4 | 4096 | Prefill | FP16 | **1294.84** | – | – | – |
| | | | NSNQuant-2b | 1761.45 | 104.13 | 14.69 | 324.16 |
| | | Decode (per step) | FP16 | 49.34 | – | – | – |
| | | | NSNQuant-2b | **33.17** | 0.29 | 1.70 | 0.14 |
| 32 | 512 | Prefill | FP16 | **1225.90** | – | – | – |
| | | | NSNQuant-2b | 1707.22 | 155.22 | 15.04 | 326.85 |
| | | Decode (per step) | FP16 | 50.80 | – | – | – |
| | | | NSNQuant-2b | **36.49** | 0.34 | 1.88 | 0.66 |
| 128 | 128 | Prefill | FP16 | **1223.20** | – | – | – |
| | | | NSNQuant-2b | 1698.50 | 105.76 | 15.02 | 329.62 |
| | | Decode (per step) | FP16 | 63.45 | – | – | – |
| | | | NSNQuant-2b | **45.25** | 0.83 | 1.87 | 2.43 |

Table 22: Evaluation results on AIME-2024 with DeepSeek-R1-Distill-Llama-8B

| Method | pass@1 |
|---|---|
| FP16 | 43.3 |
| CQ-4c9b | 40.7 |
| NSNQuant-2b | 40.0 |
| CQ-8c10b | 6.7 |
| NSNQuant-1b | 16.7 |

are not suitable for low-bit quantization. Their evaluation results on GSM8K, HumanEval, CoQA, and MMLU with LLaMA3.1-8B-Instruct are shown in Table 23. We use a clipping ratio of 0.95, following QuaRot, and use a group size of 128 for both keys and values. We only train R2 rotation matrix in SpinQuant, since we don't apply quantization to weights or activations. We also apply the residual policy of NSNQuant, following the setting in the main experiments. The result shows that QuaRot and SpinQuant underperform in every dataset, and they are worse than our weakest baseline, KIVI-2.

Table 23: Evaluation results on GSM8K, HumanEval, CoQA, and MMLU with LLaMA3.1-8B-Instruct.

| Method | Bits | GSM8K (8-shot, CoT) | HumanEval | CoQA | MMLU (4-shot, CoT) | | | |
|---|---|---|---|---|---|---|---|---|
| | | | | | Humanities | STEM | Social | Other |
| KIVI-2 | 2.38 | 64.59 | 48.17 | 63.60 | 64.44 | 50.09 | 66.84 | 66.1 |
| QuaRot (W16A16KV2) | 2.25 | 48.67 | 47.56 | **64.05** | 57.32 | 41.91 | 62.64 | 60.51 |
| SpinQuant (W16A16KV2) | 2.25 | 41.09 | 40.24 | 62.52 | 56.30 | 39.53 | 59.85 | 59.39 |
| NSNQuant-2b | 2.23 | **75.89** | **56.10** | 63.83 | **71.04** | **55.64** | **73.42** | **70.74** |

# G    Visualizations

## G.1    Inter-channel correlation after the NSN and Hadamard

We verify that the NSN and Hadamard transform align each channel distribution with the standard normal distribution. However, large inter-channel dependency can change its multi-dimensional shapes, affecting the efficiency of our tuned codebooks. Figure 10 shows the layer-wise mean absolute correlation (MAC) between channels after the NSN and Hadamard transform. Using WikiText-2 and LLaMA2-7B model, we measure the correlations between different channels which are grouped and quantized together. The results show that the correlation is generally small except for the early

layers. This indicates our codebook is close to optimal in the later layers, while there is still room for improvement in the early layers.

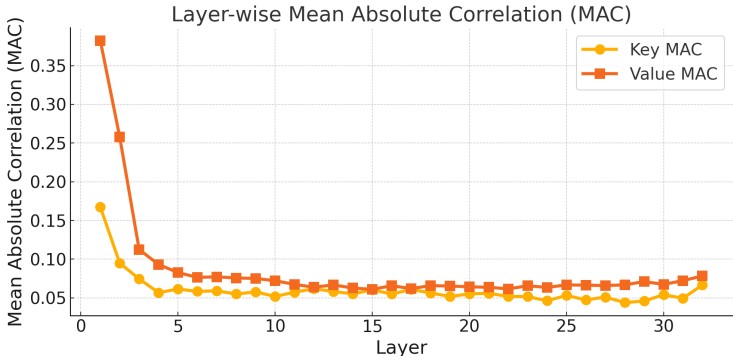

Figure 10: Layer-wise mean absolute correlation (MAC) between different channels. The results are measured on WikiText-2 with LLaMA2-7B.

## G.2 t-SNE visualization of keys and values

To visualize the divergent token distribution of different datasets, we plot 2D t-SNE results in Figure 11. For MultiNews, LCC, and SAMSum, the "context" column is used for visualization. We sample first 10 samples in the test split for each datasets. The input length is limited to 2048 tokens, resulting in ∼20000 tokens per dataset. We run t-SNE for all tokens, and randomly select 2000 tokens from each dataset for plotting. None of the normalization methods or rotations are applied to keys and values in this visualization and pre-RoPE keys are used for visualization.

## G.3 Channel-wise mean and standard deviation

We visualize the channel-wise mean and standard deviation of keys and values after the NSN and Hadamard transform in Figure 12 and 13. Although mean and standard deviation are generally close to 0 and 1 as expected, there are non-negligible errors in some of the heads in the early layers.

# H Comparison with CQ

**Motivation**   Although CQ and NSNQuant both quantize multiple channels jointly, their motivations behind adopting vector quantization (VQ) differ. CQ employs VQ to capture correlations across channels, whereas NSNQuant is inspired by the observation from QuIP# [37] that VQ performs particularly well when quantizing ball-shaped vectors. Since the channel distribution is aligned with the standard normal distribution regardless of model and data, we can build an optimized codebook for compressing the normal distribution data in advance.

**Calibration**   CQ requires calibration data and backward passes through model weights to compute Fisher information. In contrast, NSNQuant does not require any external data, and its codebook tuning can be completed within a few minutes on a single GPU. Moreover, the tuned codebook is model-agnostic: it can be reused across models as long as the hidden dimension per head remains unchanged. This property makes NSNQuant much easier to integrate and significantly improves its generalization ability.

**Codebook**   CQ requires a separate codebook for each group of coupled channels. For example, in CQ-8c10b, the total size of all codebooks matches the KV cache for 1024 tokens in fp16, which is equivalent to 16384 tokens in 1-bit representation. In contrast, NSNQuant uses a single shared codebook for the entire model, resulting in almost no additional memory overhead.

# I  Limitation and broader impacts

## I.1  Limitation

Although proposed Normalize-Shift-Normalize (NSN) is empirically proven to align the output distribution with the standard normal distribution, we find that it does not work well in the early layers due to outlier channels with huge variances. Our analysis shows that the quantization errors in these layers still remain low, but handling these channels properly will further improve performance. Also, NSNQuant does not consider inter-channel correlations or dependencies, while CQ induces errors by relying too much on them. Exploring a middle ground between two methods will be beneficial: exploiting inter-channel relations, while avoiding overfitting to calibration datasets.

## I.2  Broader impacts

Our work improves the scalability and efficiency of LLM inference by reducing memory usage and increasing throughput. This enables broader adoption of LLMs across diverse applications and hardware setups. By significantly reducing memory consumption for long-context inference, our method makes advanced use cases like long-context reasoning feasible even on devices with constrained memory resources.

That said, our work does not provide any empirical analysis regarding safety aspects. As a result, applying quantization may introduce unintended effects such as hallucinations or degraded reliability, especially in sensitive scenarios. We therefore advise caution when deploying quantized models in safety-critical applications.

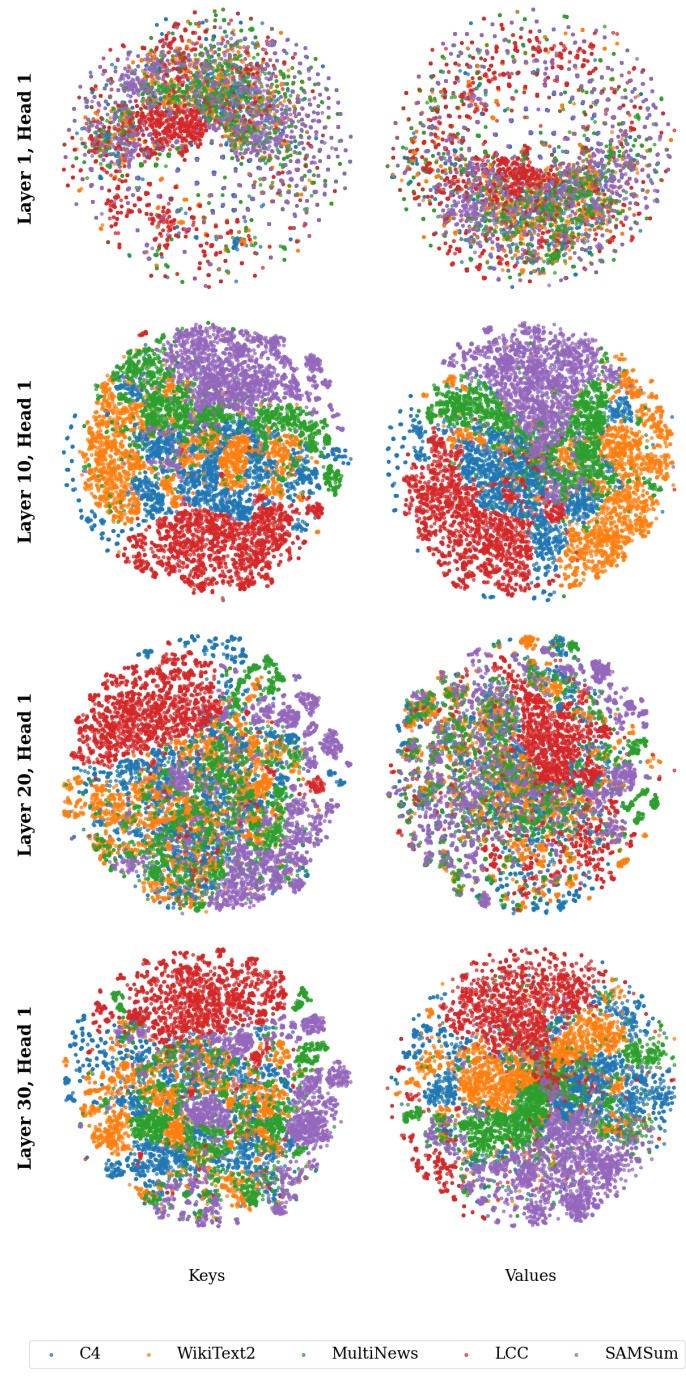

Figure 11: t-SNE visualization LLaMA3.1-8B key and value.

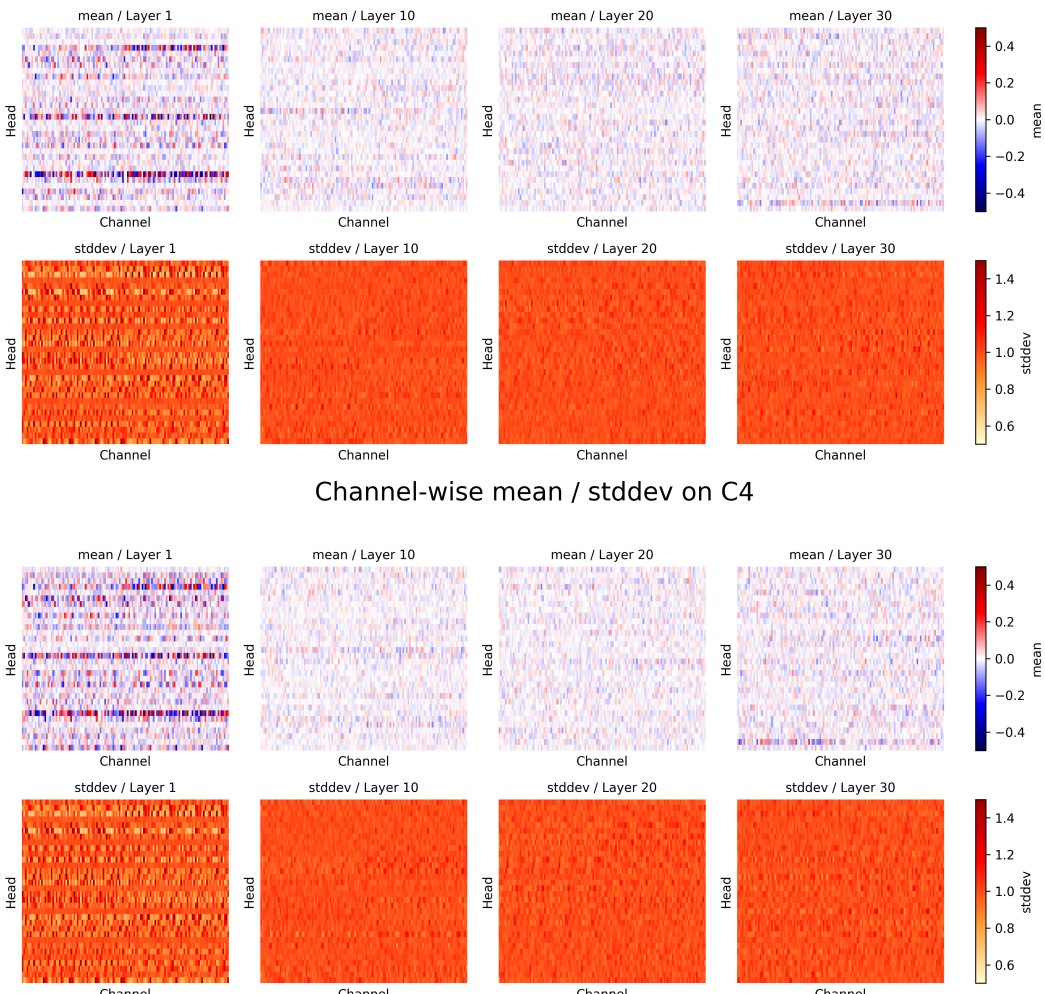

Figure 12: Visualizations of channel-wise mean and standard deviation of **keys** after applying NSN and Hadamard transform to LLaMA2-7B on WikiText-2 (top) and C4 (bottom). The first sample from the test split of each dataset is used for visualization. While NSN overall performs standardization fairly well, it struggles in certain heads of the early layers.

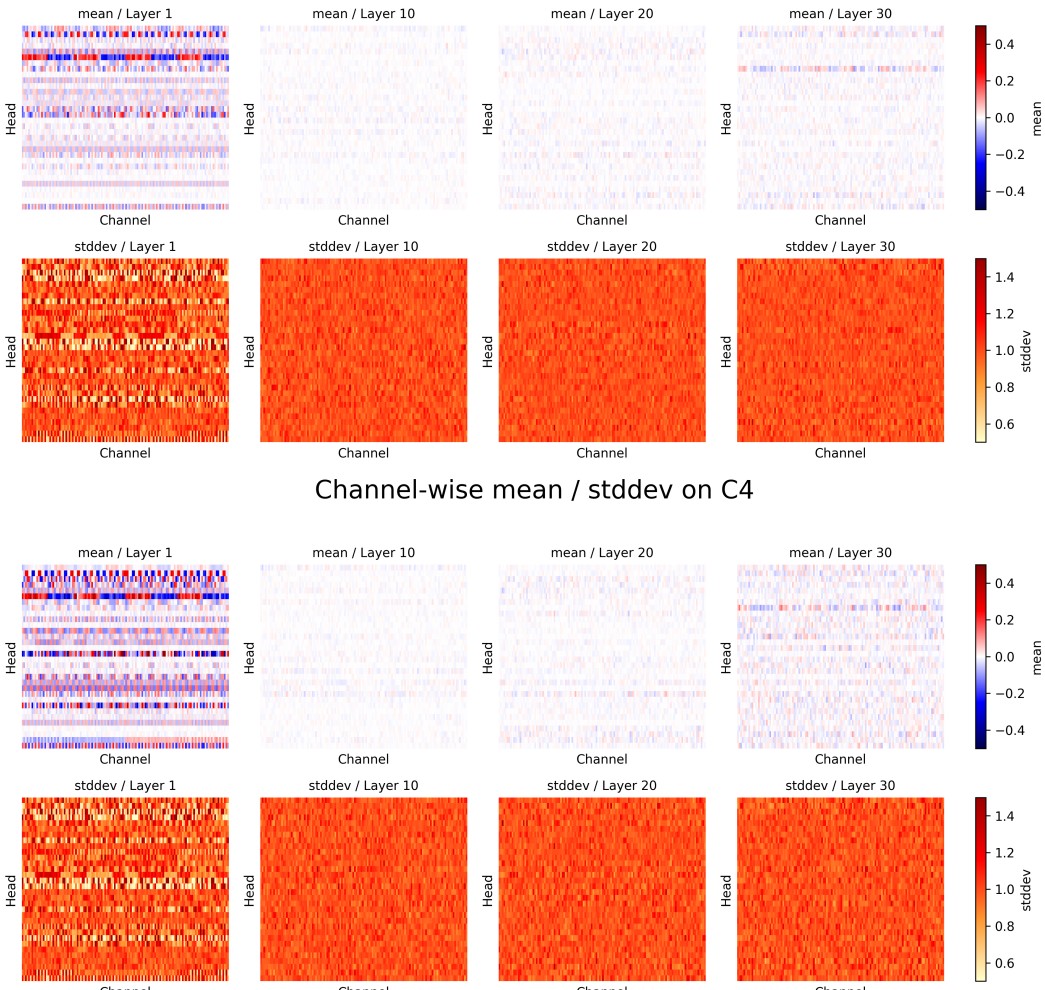

Figure 13: Visualizations of channel-wise mean and standard deviation of **values** after applying NSN and Hadamard transform to LLaMA2-7B on WikiText-2 (top) and C4 (bottom). The first sample from the test split of each dataset is used for visualization. While NSN overall performs standardization fairly well, it struggles in certain heads of the early layers.

