# OpenReview forum: "NSNQuant: A Double Normalization Approach for Calibration-Free Low-Bit Vector Quantization of KV Cache"
_NeurIPS.cc/2025/Conference — NeurIPS 2025 poster_

### Official Review · Reviewer_K1rM · 2025-06-17

**Clarity:** 2
**Significance:** 3
**Originality:** 3
**Rating:** 4
**Confidence:** 3

**Summary:**

In this study, we first identify the problem of distribution shift in current KV cache vector quantization, and then propose a new vector quantization method called “NSNQuant” that is robust to distribution shift in datasets: normalization and mean shift followed by Hadamard Transformation to approximate standard normal distribution. Experiments have demonstrated that NSNQuant consistently outperforms prior methods in both 1-bit and 2-bit settings.

**Questions:**

I would like to point out some readability issues which directly affects our understanding of the proposed method.
- Figure 2: I could not understand the pipeline of the overall structure. Firstly, W_q, W_k, and W_v are model parameters related with attention? or is it a latent representation? Secondly, what is R_Had? Thirdly, what is the mathematical formula before and after softmax? Please explain the detail in natural language.
- Normalization alongside token dimension (in NSN procedure): I think it is NOT possible when we start inference because in the beginning of the inference, we cannot foresee the future tokens. Please explain the detailed procedure.
- "1-bit and 2-bit settings": I'm not quite sure which part is quantized in the proposal. Is it that each vector element in the codebook is quantized to 1-bit or 2-bit?
- Figure 3: What is channel distribution? Does it refer to the distribution of values in a single selected channel?

**Ethical Concerns:**

["NO or VERY MINOR ethics concerns only"]

**Final Justification:**

Our remaining concerns are clarified by the author's detailed explanations.

So, I have raised my score from borderline reject to borderline accept.

I hope the authors will incorporate the information in the revised manuscript version.

**Limitations:**

Yes.

**Paper Formatting Concerns:**

No.

**Quality:**

3

**Strengths And Weaknesses:**

Strength:
- Based on experimental results, the author points out the problems with existing research and argues for the need for new methods.
- The novelty of the proposed method is directly linked to solving the problem of distributional shift in KV cache vector: normalization and mean shift followed by Hadamard Transformation to approximate standard normal distribution.
- The effectiveness of the proposed method is demonstrated by conducting comparative experiments with representative quantization baselines.
- Source code is available for reproducibility.

Weakness:
- Readability: some of the proposed method is hard to understand. See the details for "Questions".

---

> ### Author Rebuttal · Authors · 2025-07-31
>
> We sincerely appreciate the time and effort you devoted to reviewing our submission.
> We acknowledge that some notations of our method section can be misleading and make it hard to understand the paper.
> We will polish the method section and fix informal notations and expressions based on your thoughtful reviews.
>
> **[Q1] Clarification on Figure 2**
> - **W_q, W_k, and W_v** are the weights of the linear projection layers for query, key, and value, respectively. In HuggingFace Transformers, these correspond to the weights of `q_proj`, `k_proj`, and `v_proj` in the `LlamaAttention` module. We omitted detailed explanation of these notations since they are widely adopted in prior works such as LoRA [1] and SpinQuant [2].
> - **R_Had** refers to the **Hadamard matrix**, and it is used in Figure 2 to visualize the Hadamard transformation. Since applying a Hadamard transform is equivalent to multiplying by a Hadamard matrix, many previous works (e.g., SpinQuant [2], RotateKV [3]) describe this as applying a rotation, using the notation **R** and calling it "online rotation." In our case, we use this notation to indicate that a Hadamard transform is applied in the pipeline. This transformation is linear and can be merged into the projection weights (e.g., `W_v`, `W_o`) to minimize computational overhead, as shown in the figure. We also use the notation **q_Had** to denote the Hadamard-transformed query vector. We acknowledge that these notations might be confusing and will add a clarifying note in the figure caption in the final version.
> - The mathematical formula before and after softmax is explained in Appendix B, as stated in the caption. We explain how each formula is derived using the restoration formula v = s_1(s_2 v_nsn + o). Due to the page limit, we placed this derivation in the appendix, but we will consider including it in the main text in the camera-ready version to improve clarity.
>
> ---
>
> **[Q2] Feability of normalization in NSN**
>
> NSN procedure incorporates two types of transformation:
> 1. Token-wise normalization: normalize each key or value token to have a scale of sqrt(d)
> 2. Channel-wise centering: make each channel dimension have a mean of 0
>
> The token-wise normalization is feasible because norm of each token can be easily computed online.
> However, as you pointed out, channel-wise centering is infeasible because it requires statistics of future tokens.
>
> To resolve this issue, we adopt a concept of residual from KIVI [3], as explained in the Section 3.2.
> We split KV cache into two parts: one part with quantized cache and the other part (residual) with full-precision cache for recent tokens.
> Once the size of residual (which is set to 64 in this paper) reaches its maximum capacity, the tokens within the residual are transformed together through NSN and Hadamard transform, and then quantized using a codebook.
> During these steps, channel-wise mean is now calculable since multiple tokens are quantized at once.
> KIVI also adopts residual for a similar reason: they quantize key cache along the channel dimension.
>
> We omitted residual from Figure 2 for simplicity of the figure.
> However, we acknowledge that it can be misleading and will try to visualize the residual in the Figure 2 as well.
>
> ---
>
> **[Q3] Regarding the term "1-bit setting" and "2-bit setting"**
>
> We used the terms **"1-bit setting"** and **"2-bit setting"** to refer to the setting where KV cache is quantized into approximately 1 and 2 bits per each value, respectively. In "k-bit setting", "k-bit quantization methods" are adopted for quantizing KV cache. This categorization is important because average bit width (average bit per value) decides the memory consumption.
> Specifically:
>
> - Methods with an average bit width around 1 (e.g., **NSNQuant-1b**, **CQ-8c10b**, **KVQuant-1b**) are categorized as **1-bit quantization methods**.
> - Likewise, methods with an average bit width around 2 (e.g., **NSNQuant-2b**, **CQ-4c9b**, **KVQuant-2b**) are categorized as **2-bit quantization methods**.
>
> These methods compress both **key** and **value** caches into the corresponding bitwidth.
> For example, in **NSNQuant-1b**, each key or value token is divided into multiple **8-dimensional sub-vectors**, and each subvector is quantized using **8-bit indices** that point to entries in a **256-entry codebook** (each entry being an 8D vector).
> Since **8 bits are used to represent each 8D vector**, this results in an **average of 1 bit per dimension** (i.e., 8 bits / 8 dimensions = 1).
> Note that the average bit width for NSNQuant-1b is slightly larger than 1 because of additional parameters incurred by NSN.
>
> ---
>
> **[Q4] Regarding the term "channel distribution" (Figure 3)**
>
> Yes, we use the expression "channel distribution" to denote the distribution of values in a single selected channel.
> We acknowledge that the term may be ambiguous, and we will revise it to a clearer phrase like "per-channel value distribution" in the final version.
>
> ---
>
> We sincerely thank the reviewer for pointing out the readability issues.
> Your feedback is valuable and will help us significantly improve the clarity of the paper.
> We truly appreciate the opportunity to further refine our work based on your suggestions.
> We hope these answers help you understand the paper in depth.
> If you have any additional questions or need further clarification, we would be glad to provide more details.
>
> ---
> **References**
> [1] Lora: Low-rank adaptation of large language models, Hu, Edward J., et al., ICLR 2022
> [2] Spinquant: Llm quantization with learned rotations, Liu, Zechun, et al., ICLR 2025.
> [3] KIVI: A Tuning-Free Asymmetric 2bit Quantization for KV Cache, Liu, Zirui, et al., ICML 2024

---

> > ### Comment · Reviewer_K1rM · 2025-08-04
> >
> > Thank you for the author's reply.
> >
> > Our remaining concerns are clarified by the author's detailed explanations.
> >
> > I hope the authors will incorporate the information in the revised manuscript version.

---

### Official Review · Reviewer_C5ZW · 2025-06-30

**Clarity:** 3
**Significance:** 2
**Originality:** 2
**Rating:** 4
**Confidence:** 4

**Summary:**

It is a well-known problem that memory issue is the major bottleneck of LLM (Large Language Model) inference because LLM inference is memory-bounded. Recently, a lot of work has emerged to resolve this issue, and one of the representative solutions is KV cache quantization.

This work points out that the distribution of the KV cache differs according to inputs, thus the distributions of the KV cache shift depending on the calibration set. It leads to performance degradation in general areas. To resolve this distribution shift problem, this work proposes token-wise normalization, following channel-wise centering and token-wise normalization again. With the above process, the distributions of tokens become standard normal distributions, and it prevents distribution shifts of the KV cache after quantization. Also, by applying Hadamard transform and scale adjustment, the paper achieves state-of-the-art results among other KV cache compression works.

**Questions:**

- How much is the overhead of NSNQuant and Hadamard transform? The paper compares memory consumption and throughput with 16-bit results. Is the overhead negligible compared to previous KV cache quantization works? Or, those proposals are harmful to the overall memory consumption or throughput?

**Ethical Concerns:**

["NO or VERY MINOR ethics concerns only"]

**Final Justification:**

Under the assumption that the reviewers incorporates the reviewers’ comments in the paper, I think that the paper is worth accepting.

**Limitations:**

Yes

**Paper Formatting Concerns:**

There isn't any formatting concerns

**Quality:**

3

**Strengths And Weaknesses:**

Strengths
- This work shows the distribution shift problem that occurred while applying vector quantization to the KV cache.
- This work resolves the above problem in a simple yet effective way; add token-wise normalization, channel-wise shift, and token-wise normalization to the KV cache, along with Hadamard transform.
- This work provides optimized kernels to execute vector quantization, scale adjustment, and attention with quantized KV caches.

Weaknesses
- Along with the proposed method, NSNQuant and Hadamard transform, this paper applies other techniques such as scale adjustment and double quantization. However, the effect of those suggestions is not addressed in the experiments.
- Llama and Mistral have similar architectural features (e.g., attention architecture, activation function, and normalization function). The paper should have examined other LLM models that have different architectural features to verify the general effectiveness of the proposal.

---

> ### Author Rebuttal · Authors · 2025-07-31
>
> We sincerely appreciate the time and effort you devoted to reviewing our submission.
>
> **[W1] Missing analysis for scale adjustment and double quantization**
>
> We've already included the analysis for different components of our method in Appendix C.
> Please refer to Appendices C.3 and C.6 for the ablation studies on scale adjustment and double quantization.
>
> ---
>
> **[W2] Experiments on more architectures**
>
> As you pointed out, LLaMA and Mistral share similar architectural features. Therefore, we add experiments with Qwen1.5-MoE-A2.7B and Qwen1.5-MoE-A2.7B-Chat models to verify that our method is effective even with the modern MoE (Mixture-of-Experts) architecture.
>
> [1] PPL evaluation result on WikiText-2 and C4 (Qwen1.5-MoE-A2.7B)
>
> | Method         | WikiText-2 | C4     |
> |----------------|-----------|--------|
> | FP16           | 6.79      | 8.88   |
> | NSNQuant-2b    | **7.07**      | **9.31**   |
> | CQ-4c9b        | 7.11      | 9.52   |
> | NSNQuant-1b    | 9.93      | 13.05  |
> | CQ-8c10b       | **8.46**      | **12.41**  |
>
> [2] Evaluation results on HumanEval, GSM8K-CoT, and CoQA (Qwen1.5-MoE-A2.7B-Chat)
> | Method         | HumanEval | GSM8K (8-shot, CoT) | CoQA  |
> |----------------|----------------|--------------------------|----------|
> | FP16           | 29.27          | 52.01                    | 59.62    |
> | NSNQuant-2b    | **26.22**          | **46.78**                    | **59.03**    |
> | CQ-4c9b        | 24.39          | 43.97                    | 58.62    |
> | NSNQuant-1b    | **17.68**          | **23.88**                    | **55.57**    |
> | CQ-8c10b       | 12.20          | 19.26                    | 50.32    |
>
>
> Although NSNQuant-1b achieves slightly worse PPL compared to CQ-8c10b, it achieves much better scores on lm-eval tasks. These results reinforce the claim that NSNQuant is the most effective method across different datasets and models.
>
> ---
>
> **[Q1] Latency of NSN, Hadamard, and VQ**
>
>
> To investigate the overhead of each component of NSNQuant, we measure the latency caused by each component: NSN, Hadamard transform and VQ.
> Here, the double quantization is included as part of NSN, and the scale adjustment is grouped under VQ.
> We evaluate latency under two different settings with LLaMA2-7B, each averaged over 100 runs to ensure stable measurements.
> We report per-token latency for the decoding stage.
> All experiments are done on a linux server with a single A100-80GB GPU.
>
> **[1]** Setting 1: batch size = 1, prompt length = 2048, generated tokens = 128
>
> This setting is for comparison with CQ. In the CQ paper, authors provide latency measurements on Appendix K. They used the batch size of 1, prompt length of 2000 and generate 100 tokens.
> We try to match this setting as closely as possible, and set the closest numbers which are divisible by residual size (64).
> Since the experiments are conducted with different GPUs (CQ: A100-40GB, ours: A100-80GB), it's hard to directly compare these results.
> We instead compare the **relative overhead** added over the FP16 baseline, as the CQ inference kernel is not publicly available.
>
> **[1-1]** Prefill
> | Method         | Total | NSN | Hadamard transform | VQ |
> |----------------|-----------|--------|-------|-----|
> | FP16           | 169.85ms | N/A | N/A | N/A |
> | NSNQuant-2b    | 244.73ms | 22.51ms | 2.97ms | 42.22ms |
>
> **[1-2]** Decode (per token)
> | Method         | Total | NSN | Hadamard transform | VQ |
> |----------------|-----------|--------|-------|-----|
> | FP16           | 28.21ms | N/A | N/A | N/A |
> | NSNQuant-2b    | 33.95ms | 0.30ms | 1.83ms | 0.08ms |
>
> The result shows that NSNQuant-2b requires 44% and 20% of additional overhead in prefill and decode, respectively, compared to the full precision baseline.
> Despite the additional latency from NSN and Hadamard transforms, the overhead remains lower than CQ (see Table 14 in the CQ paper).
> CQ-4c8b requires nearly twice the prefilling time and incurs a 26% overhead in the decoding stage.
>
> We attribute this result to the optimized custom kernel implementation.
> In fact, our VQ implementation can be used in K-Means algorithm, and this can boost the K-Means algorithm needed for CQ.
> Surprisingly, using our implementation, the centroid for CQ-8c10b can be found in 15 minutes with a single RTX-3090, whereas the original CQ paper reports that it requires more than 100 minutes.
>
>
> **[2]** Setting 2: batch Size = 32, prompt length = 512, generated tokens = 64
>
> This setting reflects a more realistic deployment scenario, using a moderate batch size and prompt length.
>
> **[2-1]** Prefill
> | Method         | Total | NSN | Hadamard transform | VQ |
> |----------------|-----------|--------|-------|-----|
> | FP16           | 1225.90ms | N/A | N/A | N/A |
> | NSNQuant-2b    | 1707.22ms | 155.22ms | 15.04ms | 326.85ms |
>
> **[2-2]** Decode (per token)
> | Method         | Total | NSN | Hadamard transform | VQ |
> |----------------|-----------|--------|-------|-----|
> | FP16           | 50.80ms | N/A | N/A | N/A |
> | NSNQuant-2b    | 36.49ms | 0.34ms | 1.88ms | 0.66ms |
>
>
> While NSNQuant-2b has higher prefilling latency, it achieves lower decoding latency due to reduced DRAM access during attention computation.
>
> Interestingly, both settings show that while VQ has the largest overhead in the prefill stage, Hadamard transform has the largest overhead in the decoding stage. This is similar to MLP layers becoming the main bottleneck in the decoding stage where it becomes memory-bound.

---

> > ### Comment · Reviewer_C5ZW · 2025-08-04
> >
> > The reviewer think that it takes a too long time to run LLMs with the proposed method, considering that the quantized bit is much lower than FP16. Also, even though we don't consider the additional overheads from NSN, Hadamard transform, and VQ, the inference time of NSNQuant is longer than that of FP16 in setting 1.
> >
> > However, the proposed method can help reduce decoding latency with a large batch size (setting 2; a more realistic scenario to serve LLMs). With the additional advantages induced by using low KV Cache, such as low memory occupancy, the proposed method can help reduce latency and cost for serving LLMs.
> >
> > Therefore, the review decide to keep the rating of acceptance.

---

### Official Review · Reviewer_Ynez · 2025-06-30

**Clarity:** 3
**Significance:** 3
**Originality:** 3
**Rating:** 4
**Confidence:** 3

**Summary:**

This paper proposes NSNQuant, a novel calibration-free vector quantization method for KV Cache. Its key innovation is the Normalization-Shift-Normalization module, which aligns channel distributions with the standard normal distribution. This approach addresses a key limitation of prior methods, the reliance on specific calibration datasets, which limits generalizability. The method demonstrates superior accuracy and efficiency compared to existing techniques through comprehensive evaluation.

**Questions:**

1. Why is the order of applying NSN different for keys and values? Is there a specific reason or theoretical analysis supporting this design choice?
2. Which dataset is used for codebook fine-tuning? How does the choice of dataset impact performance and generalization ability?
3. Since CQ was calibrated on WikiText-2, why does it still underperform compared to NSNQuant?

**Ethical Concerns:**

["NO or VERY MINOR ethics concerns only"]

**Final Justification:**

Most of my concerns have been addressed by authors' response.
I do not raise the score because the incurred overhead is relatively high, which may have a negative impact on practical usage.
However, this paper still deserves an acceptance due to its contributions and evaluation results compared to related work.
So, we choose to maintain the original score -- borderline accept.

**Limitations:**

Yes

**Paper Formatting Concerns:**

No formatting issues

**Quality:**

3

**Strengths And Weaknesses:**

Strengths:

1. The paper introduces a novel calibration-free vector quantization method. It incorporates the NSN module to align channel distributions with the standard normal distribution. This is a significant innovation addressing a core limitation of existing methods that depend on calibration datasets.

2. The paper provides a comprehensive evaluation of the proposed method across various datasets, demonstrating its superior performance over previous methods. It also includes a well-structured ablation study validating the effectiveness of its components.

3. The paper proves the mathematical effectiveness of the proposed module. It also provides a CUDA implementation to enhance inference efficiency.

Weaknesses:

1. This method is sensitivite to the choice of hyperparameters, e.g., residual size when quantization bit is 1.
2. This method incur extra latency due to the additional normalization steps especially in prefilling stage and the paper does not provide a detailed analysis of its latency impact.

---

> ### Author Rebuttal · Authors · 2025-07-31
>
> We sincerely appreciate the time and effort you devoted to reviewing our submission.
>
> **[W1] Sensitivity to hyperparameter selection**
>
> While NSNQuant-1b appears sensitive to the choice of residual size, we emphasize that this sensitivity is not due to random fluctuations but reflects a controllable trade-off between performance and memory cost. As residual size increases, performance improves due to the inclusion of more full-precision tokens, at the cost of additional memory usage.
> We believe that this cost-reliability tradeoff will become less pronounced as future work improves the quantization quality further.
> Please refer to Appendix C.2 for detailed results.
>
> Furthermore, we would like to clarify that NSNQuant is not broadly sensitive to hyperparameter selection. In practice, residual size is the only major hyperparameter that significantly affects performance. Other components, including the NSN process and the codebook, are fixed and do not require tuning across different models or tasks.
>
> ---
>
> **[W2] Latency of NSN, Hadamard transform and VQ**
>
>
> To investigate the overhead of each component of NSNQuant, we measure the latency caused by each component: NSN, Hadamard transform and VQ.
> Here, the double quantization is included as part of NSN, and the scale adjustment is grouped under VQ.
> We evaluate latency under two different settings with LLaMA2-7B, each averaged over 100 runs to ensure stable measurements.
> We report per-token latency for the decoding stage.
> All experiments are done on a linux server with a single A100-80GB GPU.
>
> **[1]** Setting 1: batch size = 1, prompt length = 2048, generated tokens = 128
>
> This setting is for comparison with CQ. In the CQ paper, authors provide latency measurements on Appendix K. They used the batch size of 1, prompt length of 2000 and generate 100 tokens.
> We try to match this setting as closely as possible, and set the closest numbers which are divisible by residual size (64).
> Since the experiments are conducted with different GPUs (CQ: A100-40GB, ours: A100-80GB), it's hard to directly compare these results.
> We instead compare the **relative overhead** added over the FP16 baseline, as the CQ inference kernel is not publicly available.
>
> **[1-1]** Prefill
> | Method         | Total | NSN | Hadamard transform | VQ |
> |----------------|-----------|--------|-------|-----|
> | FP16           | 169.85ms | N/A | N/A | N/A |
> | NSNQuant-2b    | 244.73ms | 22.51ms | 2.97ms | 42.22ms |
>
> **[1-2]** Decode (per token)
> | Method         | Total | NSN | Hadamard transform | VQ |
> |----------------|-----------|--------|-------|-----|
> | FP16           | 28.21ms | N/A | N/A | N/A |
> | NSNQuant-2b    | 33.95ms | 0.30ms | 1.83ms | 0.08ms |
>
> The result shows that NSNQuant-2b requires 44% and 20% of additional overhead in prefill and decode, respectively, compared to the full precision baseline.
> Despite the additional latency from NSN and Hadamard transforms, the overhead remains lower than CQ (see Table 14 in the CQ paper).
> CQ-4c8b requires nearly twice the prefilling time and incurs a 26% overhead in the decoding stage.
>
> We attribute this result to the optimized custom kernel implementation.
> In fact, our VQ implementation can be used in K-Means algorithm, and this can boost the K-Means algorithm needed for CQ.
> Surprisingly, using our implementation, the centroid for CQ-8c10b can be found in 15 minutes with a single RTX-3090, whereas the original CQ paper reports that it requires more than 100 minutes.
>
>
> **[2]** Setting 2: batch Size = 32, prompt length = 512, generated tokens = 64
>
> This setting reflects a more realistic deployment scenario, using a moderate batch size and prompt length.
>
> **[2-1]** Prefill
> | Method         | Total | NSN | Hadamard transform | VQ |
> |----------------|-----------|--------|-------|-----|
> | FP16           | 1225.90ms | N/A | N/A | N/A |
> | NSNQuant-2b    | 1707.22ms | 155.22ms | 15.04ms | 326.85ms |
>
> **[2-2]** Decode (per token)
> | Method         | Total | NSN | Hadamard transform | VQ |
> |----------------|-----------|--------|-------|-----|
> | FP16           | 50.80ms | N/A | N/A | N/A |
> | NSNQuant-2b    | 36.49ms | 0.34ms | 1.88ms | 0.66ms |
>
>
> While NSNQuant-2b has higher prefilling latency, it achieves lower decoding latency due to reduced DRAM access during attention computation.
>
> Interestingly, both settings show that while VQ has the largest overhead in the prefill stage, Hadamard transform has the largest overhead in the decoding stage. This is similar to MLP layers becoming the main bottleneck in the decoding stage where it becomes memory-bound.
>
> ---
>
> **[Q1] Order for applying NSN to keys**
>
> Applying NSN to values is straightforward, as they are not affected by position embedding.
> However, for keys, the presence of the RoPE (Rotary Position Embedding) requires additional consideration. Motivated by prior studies such as KVQuant and CQ, which observed that key vectors are easier to quantize before RoPE, we empirically compare applying pre-RoPE NSN and post-RoPE NSN.
> As detailed in Appendix C.5, the pre-RoPE variant yields slightly better quantization quality, and we adopt this setting as our final design.
>
> ---
>
> **[Q2] Datasets used for codebook tuning**
>
> As stated in Section 3.4, we fine-tune the codebook on **synthetic standard normal data** generated by `torch.randn`.
> We do not need any external data because NSN and Hadamard transform align the KV distribution with the standard normal distribution, and it is effective across different models and datasets.
>
> Moreover, the ablation study on codebook tuning (Section 4.6) shows that the codebook tuned solely on synthetic data works well when applied to actual keys and values.
> Interestingly, the cosine similarity measured using the keys and values of LLaMA2 is very close to that measured using the synthetic data.
> This indicates that NSN successfully aligns the key-value distribution with the standard normal distribution.
>
> ---
>
> **[Q3] Lower performance of CQ on WikiText-2**
>
> Although CQ is calibrated on WikiText-2, it may still underperform compared to NSNQuant for the following reasons:
>
> - **Limited calibration set size**: CQ uses only 16 samples from WikiText-2 for calibration. This small set may fail to capture the full diversity of input distribution within the dataset, leading to suboptimal codebook generalization even on in-distribution data.
>
> - **Additional techniques in NSNQuant**: NSNQuant applies additional techniques like scale adjustment to further improve performance. Please refer to Appendix C for ablation studies on these techniques.

---

> > ### Comment · Reviewer_Ynez · 2025-08-05
> >
> > Thanks for authors' reply.
> >
> > Most of the concerns and questions have been addressed. However, the runtime overhead weakens the benefits that quantization should provide. Thus, I choose to remain the score.

---

### Official Review · Reviewer_etjS · 2025-07-02

**Clarity:** 2
**Significance:** 2
**Originality:** 3
**Rating:** 4
**Confidence:** 4

**Summary:**

The authors propose NSNQuant, a calibration-free vector quantization (VQ) method aimed at addressing the distribution-shift sensitivity observed in previous KV cache quantization methods such as CQ (Zhang et al., 2024). NSNQuant utilizes a three-step Normalize-Shift-Normalize (NSN) transformation combined with a Hadamard transform to align key-value (KV) vectors to a standard normal distribution. Consequently, this allows the codebooks to be trained solely on synthetic Gaussian data, bypassing the need for calibration datasets. Extensive experiments demonstrate that NSNQuant outperforms existing calibration-based approaches across various benchmarks.

**Questions:**

- As noted in weaknesses, how much additional latency do the proposed transformations (NSN and Hadamard) introduce during inference compared to CQ? Is the overhead negligible in practice?
- The authors mention that early layers exhibit higher variance after transformation. What might be the possible causes for this phenomenon? Is there a risk that quantization errors in these early layers propagate and compound across subsequent layers?
- In the case of CQ, would scaling up the calibration set help reduce out-of-distribution errors? While NSNQuant benefits from faster training of the centroids, KV cache applications prioritize their inference efficiency. In such scenarios, how does NSNQuant compare to CQ if calibration can be scaled effectively?
- VQ is powerful for compression due to its strong representation capacity. While enforcing a prior enables calibration-free codebook training, could this constraint limit VQ’s expressiveness? For example, for the early layers, would finetuning the codebook with calibration sets further mitigate the quantization error?

**Ethical Concerns:**

["NO or VERY MINOR ethics concerns only"]

**Final Justification:**

The authors have provided a justification for the method’s latency, which addressed my concerns. Given its unique advantages over the baseline, I will maintain my score of acceptance.

**Limitations:**

The authors have discussed limitations of their study properly in the appendix.

**Quality:**

3

**Strengths And Weaknesses:**

Strengths:
- The proposed idea of aligning KV states to a known distribution prior is novel and can effectively removes the calibration requirement.
- The authors show a variance bound on the transformed key/value states in Lemma 1.
- Experiments are comprehensive, clearly illustrating NSNQuant’s superiority over existing quantization baselines, particularly its robustness to distribution shifts which notably improves upon CQ.

Weaknesses:
- Efficiency evaluation is inadequate. Since NSNQuant involves additional computational steps (NSN and Hadamard transformations) during inference, it is essential to measure and clearly report the latency overhead. The absence of detailed latency comparison with calibration-based methods, such as CQ, weakens the practical appeal of the proposed approach.
- The overall presentation of the paper needs improvement, e.g., Table 1 is never explicitly referenced or discussed in the main text, which is somewhat confusing.

---

> ### Author Rebuttal · Authors · 2025-07-31
>
> We sincerely appreciate the time and effort you devoted to reviewing our submission.
>
> **[W1, Q1] Latency of NSN, Hadamard and VQ**
>
> To investigate the overhead of each component of NSNQuant, we measure the latency caused by each component: NSN, Hadamard transform and VQ.
> Here, the double quantization is included as part of NSN, and the scale adjustment is grouped under VQ.
> We evaluate latency under two different settings with LLaMA2-7B, each averaged over 100 runs to ensure stable measurements.
> We report per-token latency for the decoding stage.
> All experiments are done on a linux server with a single A100-80GB GPU.
>
> **[1]** Setting 1: batch size = 1, prompt length = 2048, generated tokens = 128
>
> This setting is for comparison with CQ. In the CQ paper, authors provide latency measurements on Appendix K. They used the batch size of 1, prompt length of 2000 and generate 100 tokens.
> We try to match this setting as closely as possible, and set the closest numbers which are divisible by residual size (64).
> Since the experiments are conducted with different GPUs (CQ: A100-40GB, ours: A100-80GB), it's hard to directly compare these results.
> We instead compare the **relative overhead** added over the FP16 baseline, as the CQ inference kernel is not publicly available.
>
> **[1-1]** Prefill
> | Method         | Total | NSN | Hadamard transform | VQ |
> |----------------|-----------|--------|-------|-----|
> | FP16           | 169.85ms | N/A | N/A | N/A |
> | NSNQuant-2b    | 244.73ms | 22.51ms | 2.97ms | 42.22ms |
>
> **[1-2]** Decode (per token)
> | Method         | Total | NSN | Hadamard transform | VQ |
> |----------------|-----------|--------|-------|-----|
> | FP16           | 28.21ms | N/A | N/A | N/A |
> | NSNQuant-2b    | 33.95ms | 0.30ms | 1.83ms | 0.08ms |
>
> The result shows that NSNQuant-2b requires 44% and 20% of additional overhead in prefill and decode, respectively, compared to the full precision baseline.
> Despite the additional latency from NSN and Hadamard transforms, the overhead remains lower than CQ (see Table 14 in the CQ paper).
> CQ-4c8b requires nearly twice the prefilling time and incurs a 26% overhead in the decoding stage.
>
> We attribute this result to the optimized custom kernel implementation.
> In fact, our VQ implementation can be used in K-Means algorithm, and this can boost the K-Means algorithm needed for CQ.
> Surprisingly, using our implementation, the centroid for CQ-8c10b can be found in 15 minutes with a single RTX-3090, whereas the original CQ paper reports that it requires more than 100 minutes.
>
>
> **[2]** Setting 2: batch Size = 32, prompt length = 512, generated tokens = 64
>
> This setting reflects a more realistic deployment scenario, using a moderate batch size and prompt length.
>
> **[2-1]** Prefill
> | Method         | Total | NSN | Hadamard transform | VQ |
> |----------------|-----------|--------|-------|-----|
> | FP16           | 1225.90ms | N/A | N/A | N/A |
> | NSNQuant-2b    | 1707.22ms | 155.22ms | 15.04ms | 326.85ms |
>
> **[2-2]** Decode (per token)
> | Method         | Total | NSN | Hadamard transform | VQ |
> |----------------|-----------|--------|-------|-----|
> | FP16           | 50.80ms | N/A | N/A | N/A |
> | NSNQuant-2b    | 36.49ms | 0.34ms | 1.88ms | 0.66ms |
>
>
> While NSNQuant-2b has higher prefilling latency, it achieves lower decoding latency due to reduced DRAM access during attention computation.
>
> Interestingly, both settings show that while VQ has the largest overhead in the prefill stage, Hadamard transform has the largest overhead in the decoding stage. This is similar to MLP layers becoming the main bottleneck in the decoding stage where it becomes memory-bound.
>
> ---
>
> **[W2] Presentation of the paper**
>
> We will polish references to tables and figures, and make sure they are properly used. For example, Table 1 will be referenced in Section 3.2 to demonstrate the effectiveness of NSN. Also, we will fix typos and add missing descriptions for notations (e.g. W_q, W_k, W_v in Figure 2).
> In addition, our paper has multiple references to Appendix, which makes it hard to read the main text. This was due to the page limit, so will be improved in the final version.
>
> ---
>
> **[Q2] High covariances observed in the early layers**
>
> As mentioned in the paper, certain channels in the early layers exhibit high variance or covariance after the NSN transformation. This occurs because these channels originally have larger variances compared to the other channels, which become more pronounced through the normalization process of NSN.
>
> However, this phenomenon does not degrade quantization quality. Because of the *isotropic nature of the codebook*, NSNQuant remains robust even when such outlier channels exist: (Recall that after applying NSN and Hadamard transform, NSNQuant splits the vector into 8-dimensional subvectors and selects the closest match from the codebook.)
>
> - The codebook is trained on synthetic standard normal data (`torch.randn`), which are uniformly distributed in direction. Therefore, the quantization quality depends on the **magnitude** of the subvector rather than its **direction**.
> - Importantly, the following Hadamard transform spreads the contribution of each channel across all dimensions, making it unlikely for any single subvector to be dominated by an outlier channel. While the magnitude distribution may deviate from that of standard normal vectors, it remains within a range well-handled by the codebook.
>
> As a result, the quantization quality remains high even in the early layers as shown in Figure 5. However, the codebook is far from optimal in these layers since it does not consider correlations caused by the outlier channels. We believe that further improvements are possible by explicitly accounting for these outlier channels, and leave this for future work.
>
> ---
>
> **[Q3] Effect of calibration set size on the performance of CQ**
>
> To investigate the effect of scaling the calibration set, we test 2 variants of CQ where the size of the calibration set is doubled (From 16 to 32).
> First, we use 32 samples from WikiText-2 (W32).
> This variant enlarges the size of the calibration set.
> Second, we use 16 samples from WikiText-2 and another 16 samples from C4 (W16C16).
> This variant not only increases the size but diversifies the calibration set, covering a wider range of input distribution.
> We use LLaMA3-8B-Instruct model to test performance on the three downstream tasks: HumanEval, GSM8K, and CoQA.
> We denote two variants as W32 and W16C16, respectively.
>
> | Method         | HumanEval | GSM8K (8-shot, CoT) | CoQA  |
> |----------------|----------------|--------------------------|----------|
> | FP16           | 28.05          | 76.62                    | 61.47    |
> | NSNQuant-2b    | **32.93**          | **74.53**                    | **61.62**    |
> | CQ-4c9b        | 23.78          | 72.55                    | 60.60    |
> | CQ-4c9b (W32)       | 25.00          | 72.93                    | 60.38    |
> | CQ-4c9b (W16C16)       | 29.27          | 73.16                    | 60.98    |
> | NSNQuant-1b    | **29.27**          | **61.64**                    | **60.28**    |
> | CQ-8c10b       | 13.41          | 24.26                    | 53.38    |
> | CQ-8c10b (W32)      | 15.24          | 21.76                    | 53.37    |
> | CQ-8c10b (W16C16)      | 12.20          | 46.47                    | 54.07    |
>
> While W32 shows only marginal improvement from the baseline, W16C16 provides clearer benefits. This result shows that to improve the robustness of CQ, it is important to diversify the calibration set, rather than just scaling it up. In addition, despite improved performance, NSNQuant still shows better results. For example, on HumanEval, CQ-8c10b still suffers from significant performance degradation since its input distribution (codes) is not covered by either WikiText-2 or C4.
>
> We believe further scaling calibration set of CQ will further improve its generalization ability. However, since CQ needs gradient computation, the calibration process incurs comparable cost to fine-tuning, as the calibration set grows. Moreover, it's challenging to prepare the calibration set that adequately covers the full diversity of real-world input distributions (e.g. different languages).
> Therefore, we believe that NSNQuant remains a highly practical and robust option in general scenarios, especially when broad generalization is required.
>
> ---
>
> **[Q4] Impacts of NSN on the expressiveness of VQ**
>
> As you pointed out, applying the NSN may suppress certain channel-specific trends, potentially limiting the expressiveness of vector quantization (VQ). However, as shown in Figure 10, channel-wise correlations still remain after NSN, particularly in the early layers. This suggests that further fine-tuning the codebook using a small calibration set could indeed help reduce quantization error, especially in these layers. Moreover, NSNQuant does not consider any channel-wise  importance during codebook construction. Incorporating this information could offer an additional opportunity to enhance quantization quality.
> We believe future work could explore a promising middle ground between normalization and calibration, potentially yielding better trade-offs between generalization and expressiveness.
>
> In addition to this, we would like to emphasize that per-head codebooks or fine-tuned variants introduce additional memory overhead. For example, CQ-8c10b requires ~500MB of additional memory space with LLaMA2-7B. In contrast, NSNQuant maintains a single shared codebook across the layers, achieving strong compression and generalization without calibration.

---

> > ### Comment · Reviewer_etjS · 2025-08-04
> >
> > Thank you for the rebuttal, which has addressed my remaining concerns. I hope the authors will incorporate these results into the revised manuscript.

---

### Note · Authors · 2025-08-14

We sincerely thank the reviewers for their insightful feedback on our paper.
We are pleased that they recognized the novelty of our method (etjS, Ynez, K1rM), the comprehensiveness of our experiments (etjS, Ynez, K1rM), and the strengths of both our mathematical formulation (etjS, Ynez) and CUDA implementation (Ynez, C5ZW).
We also emphasize, as noted by reviewer C5ZW, that our method is simple yet effective, and addresses key limitations in the prior approach.

During the rebuttal period, we carefully addressed all reviewers' concerns and provided additional evidence to strengthen our claims. For example, we provided experimental results with Qwen1.5-MoE-A2.7B models to show that our method is applicable to a wide range of models. Moreover, we provided latency analyses on each component of NSNQuant with two settings: one for comparison with Coupled Quantization (CQ) [1], and one reflecting a realistic deployment scenario.

However, some reviewers still have concerns regarding the overhead of our method.
While we acknowledge that NSNQuant introduces extra computation, we believe the following points address this concern:

**[1]** The result in Setting 1 (batch size = 1, prompt length = 2048, generated tokens = 128) shows that our method (NSNQuant-2b) has lower latency overhead than CQ (CQ-4c8b).
- Despite the additional computation introduced by the NSN and Hadamard transform, our optimized VQ implementation achieves even lower overhead.
- Please note that CQ-4c8b is the weaker variant of our main competitor, CQ-4c9b, and uses a codebook half the size of CQ-4c9b.

**[2]** NSNQuant offers superior decoding performance for larger batch sizes (Setting 2), due to the reduced memory overhead.
- This is also demonstrated in Figure 4 of our paper, where NSNQuant shows much higher throughput on large batch sizes.
- In fact, Setting 1 is far from the optimal deployment configuration, as our method enables the use of large batch sizes.
- Similarly, NSNQuant is also expected to perform better with longer sequence lengths. Therefore, with long-context applications (e.g., reasoning, RAG) becoming prevalent, these performance gains grow in significance.

From these, we still believe our method provides strong efficiency benefits. We will provide a more detailed explanation in the revised version.

---

**References**
[1] KV Cache is 1 Bit Per Channel: Efficient Large Language Model Inference with Coupled Quantization, Zhang, Tianyi, et al., NeurIPS 2024

---

### Decision · Program_Chairs · 2025-09-17

**Decision:**

Accept (poster)

**Comment:**

The paper studies the problem of KV cache compression using vector quantization (VQ) techniques. It first makes an important observation that the VQ techniques that rely on calibration data to learn the codebook don't generalize well to OOD data. To address this, the paper first transforms the keys, and values to an approximately Gaussian distribution. It then relies on a codebook learned for Gaussian data to compress the transformed keys, and values. This technique is calibration data free and doesn't face the OOD issues faced by traditional VQ techniques that  rely on calibration data.  Experimental results show that the proposed technique generalizes well to diverse datasets.

All the reviewers appreciated the novelty in the work, and the thorough experimental evaluation, and are inclined towards accepting it. However, there are some key limitations of the proposed technique which need to be addressed in the paper.
 - We ideally want a single Hadamard matrix to transform all our data into approximate Gaussian distribution. It's not clear if this is even feasible. (Lemma 1 doesn't address this because it considers a random Hadamard transform.)  Intuitively, one can always construct data points where a given Hadamard transformation can fail to yield gaussian distribution. If that happens quite frequently, the proposed technique can break too. Unfortunately, the experiments in the paper don't address this. They are conduced only on two datasets (wikitext and C4). It would be good if the authors show how the performance varies on diverse datasets.
 - As pointed out by multiple reviewers, the additional transformations performed on keys, values increase the latency of the proposed technique. While the authors addressed this to some extent in their rebuttal, I would prefer having a thorough discussion on this in the main paper (including long context latency evals).

I urge the authors to incorporate this feedback into the final camera ready version.